# Nanomaterial-Based Electrochemical Nanodiagnostics for Human and Gut Metabolites Diagnostics: Recent Advances and Challenges

**DOI:** 10.3390/bios12090733

**Published:** 2022-09-06

**Authors:** Amit K. Yadav, Damini Verma, Reena K. Sajwan, Mrinal Poddar, Sumit K. Yadav, Awadhesh Kumar Verma, Pratima R. Solanki

**Affiliations:** 1Nano-Bio Laboratory, Special Centre for Nanoscience, Jawaharlal Nehru University, New Delhi 110067, India; 2Department of Biotechnology, Vinoba Bhave University, Hazaribagh 825301, India

**Keywords:** biosensors, biomarkers, diagnosis, metabolites, nanomaterials

## Abstract

Metabolites are the intermediatory products of metabolic processes catalyzed by numerous enzymes found inside the cells. Detecting clinically relevant metabolites is important to understand their physiological and biological functions along with the evolving medical diagnostics. Rapid advances in detecting the tiny metabolites such as biomarkers that signify disease hallmarks have an immense need for high-performance identifying techniques. Low concentrations are found in biological fluids because the metabolites are difficult to dissolve in an aqueous medium. Therefore, the selective and sensitive study of metabolites as biomarkers in biological fluids is problematic. The different non-electrochemical and conventional methods need a long time of analysis, long sampling, high maintenance costs, and costly instrumentation. Hence, employing electrochemical techniques in clinical examination could efficiently meet the requirements of fully automated, inexpensive, specific, and quick means of biomarker detection. The electrochemical methods are broadly utilized in several emerging and established technologies, and electrochemical biosensors are employed to detect different metabolites. This review describes the advancement in electrochemical sensors developed for clinically associated human metabolites, including glucose, lactose, uric acid, urea, cholesterol, etc., and gut metabolites such as TMAO, TMA, and indole derivatives. Different sensing techniques are evaluated for their potential to achieve relevant degrees of multiplexing, specificity, and sensitivity limits. Moreover, we have also focused on the opportunities and remaining challenges for integrating the electrochemical sensor into the point-of-care (POC) devices.

## 1. Introduction

Nutrition conversion, which comprises a transformation from consumption of conventional to modern foods containing low nutrient diversity and high-energy density, is linked to the developed metabolic syndromes. The human diet is the composition of various elements containing non-nutrients and nutrients, providing the raw materials and driving several metabolic reactions in every body cell [1,2]. These elements and their metabolites also regulate the gene expression and functioning of cellular activities through different mechanisms. Among all the components, some of them show positive and negative effects. Researchers have studied that constant interruption in energy homeostasis or/and nutrient metabolism may occur due to the excess or deficiency of nutrients, which initiate cellular stress and lead to tissue damage and metabolic deregulations, and ultimately result in the development of metabolic syndromes [3,4,5]. Presently, it is evident that metabolism is affected by external reasons such as environment and food; internal reasons such as gene variations, gender, and age; and microbiota contact, that jointly mutate the threat of developing acquired metabolic sicknesses and various health issues in humans as shown in Figure 1. The gut microbiota has recently gained attention as it has affected human health significantly by living inside the human intestine. Accumulating evidence suggests that under normal circumstances, the gut microbiota has played a fundamental role in maintaining good human health in various aspects and also maintaining a mutualistic symbiotic relationship in which both benefit one another (eubiosis) [6,7]. Either way, the change in gut microbiota composition has shown an association with human diseases, which remains a critical area of investigation for many researchers. Various metabolites are being produced by gut microbiota from nutrients of dietary sources such as trimethylamine N-oxide (TMAO), trimethyl-amine (TMA), indole derivatives, nicotinamide, and short-chain fatty acids (SCFAs), that contribute to human diseases substantially, such as metabolic diseases and cardiovascular diseases [8,9], cancer and inflammation [10], depression [11], and colorectal cancer [12]. Some metabolites generated from the gut microbiome, including butyrate, propionate, and SCFAs such as acetate, have shown critical roles in human health and are considerably associated with kidney diseases, hypertension, and inflammatory bowel diseases [13,14]. The research in the field of gut metabolite advises that these could be used as a prognostic and diagnostic marker for numerous human diseases such as diabetes, cardiovascular, colorectal cancer (CRC), etc. [15]. The fast and cost-effective diagnosis of gut disease at the earliest stage raises challenges for patients, health institutions, and governments for efficient clinical results. It is also becoming clear that consuming diets rich in nutrient diversity and low in energy density may enhance and maintain good health.

Moreover, the measurement of metabolites has gained growing interest in clinical research. Metabolites are hardly soluble in an aqueous medium; as a result of this, the concentration of the metabolite in biological fluids is deficient. Therefore, a selective and sensitive determination of metabolites is challenging to measure with existing techniques in biological samples. Various conventional and non-electrochemical techniques such as chromatography, raman spectroscopy, surface-enhanced raman scattering (SERS) spectroscopy, and electrochemistry have been developed to analyze metabolites in biological samples [16]. SERS is a viable technology to recognize biological and chemical components through their distinct vibrational fingerprints. It opens a new way for biomedical analysis, such as point-of-care analysis, precision component-based imaging, biomolecular detection, and cancer diagnostics. SERS biosensing at the molecular level has been limited by noble metals and fixed probes [17,18]. High-performance liquid chromatography (HPLC) is a widely utilized method for metabolites detection. In HPLC, sample preparation and extraction are very important steps before the measurement occurs. Solid phase extraction (SPE) and solid phase microextraction (SPME) are conventional solvent-free pre-treatment methods associated with the HPLC. Apart from the solvent-free process, the preparation and selection of appropriate SPE and SPME sorbent are essential as they can drastically influence the ability and selectivity of sample extraction [19]. Although these methods are highly sensitive, they have the disadvantages that they are complex, time consuming, costly, and dependent on complicated instrumental setups and skilled manpower, due to which these techniques are not practical in routine measurement [20,21].

In contrast, the electrochemical techniques are more suitable for the selective determination of biomarkers. These methods are highly sensitive, cost-effective, affordable, fast, and have a simple design. Over the past years, the need has arisen in medical diagnostics fields for disposable and simple devices displaying suitable, cost-effective, user-friendly, and fast response times for mass fabrication [22,23,24]. Biosensor technologies have provided the potential to achieve the above conditions via an interdisciplinary mixture of medical science, chemistry, and nanotechnology techniques. The various advantages of biosensors are high precision, low cost, and rapid analysis applied in numerous fields, such as the fermentation industry, environmental monitoring, antibiotics detection [25,26,27], disease diagnosis [28,29,30], and medical care [31]. Biosensing technology provides significant advantages over traditional detection methods such as spectroscopy or chromatography. These advantages include reduced demand for qualified individuals, faster response time, mobility, and enhanced sensitivity. Because of the rising demand for controlling the target molecules in the human body and the environment, nanomaterials have recently sparked a lot of attention. Nanomaterials are materials with at least one dimension of 1–100 nm. Controlled tuning and synthesis of nanomaterial characteristics need data of several disciplines, including agriculture, engineering, biology, computer science, electronics, chemistry, physics, and so on, which may result in the birth of unique and multifunctional nanotechnologies. In this context, nanomaterials’ fascinating characteristics have piqued the scientific community’s interest in their use in various fields including security, food, health, information technology, and transportation, among others. It is anticipated that the intelligent usage of nanomaterials would improve the biomolecular electrical devices’ performance with high detection limitations and sensitivities.
Figure 1Factors that influence human health, extrinsic and intrinsic factors. The interaction between the host-microbiota that jointly modify the energy and nutrient metabolism, program of gene expression, and biological processes results in various health issues in humans [Adapted with permission from ref. [32]. Copyright 2015 Springer].
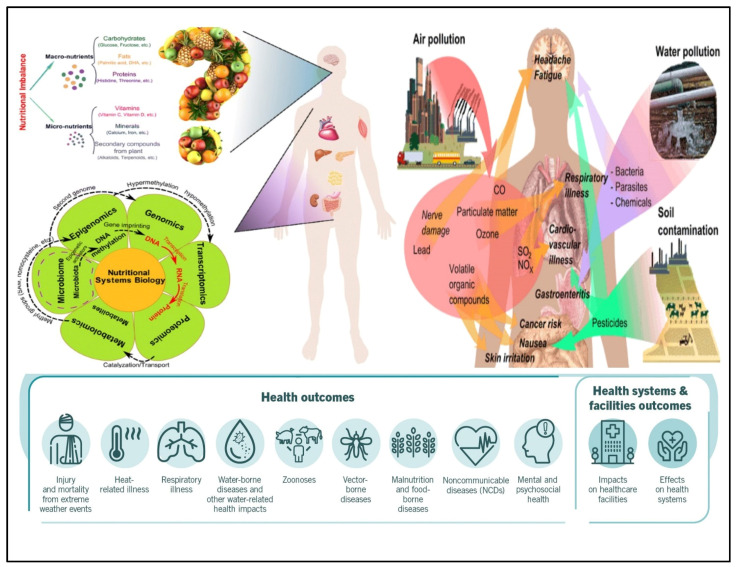


This review will acquaint the reader with recent research articles on the importance of human and gut microbiome-derived metabolites in developing disease and human health. Moreover, it highlights the current advancement and newly reported electrochemical methods that can be designed to investigate clinically appropriate human and gut metabolites for onset determination and screening of the progress of particular diseases. This is a broad and fast-growing field, and this review presents the collective and important information gathered from the past five years of research to help the researchers select appropriate biomarkers for their upcoming analysis. The current review also summarized the overviews of various sensors and their performance metrics, such as limit of detection (LOD) and dynamic range. We further highlighted the current boundaries in the biosensing field and their future encounters.

## 2. Nanomaterial-Based Biosensors for Metabolites Detection

For the determination of clinically relevant metabolites, it is crucial to understand the fundamentals of their physiological and biological functions and clinical diagnostics [33]. These molecules are associated with various biological processes, such as catalyzing and regulating biological activities, transporting small molecules, and transmitting and storing genetic information. In addition to that, they can be used as an important biomarker that helps in the reorganization and diagnosis of several diseases. Medical investigation is no longer made in clinical test centers [34]. As an alternative, it is regularly performed at numerous units, such as the hospital point-of-care unit, nonhospital take-care unit, and home [35]. The electrochemical determination methods are suitable for rapid and on-the-spot application [36]. These methods can potentially develop selective, affordable, rapid, and sensitive systems for determining various analytes, which are important for clinical diagnosis and monitoring disease treatment [37]. The electrochemical nanobiosensors have been developed by using nanomaterials in which nanomaterials act as an electrode. In electrochemical biosensors, an electrode converts a response signal into an electrical signal generated from biomolecules that act as receptors when binding to the target molecules [38,39]. In the next section of this review, we include several ways used for small molecules and ions to selectively detect (1) specific biorecognition elements such as ionophores, double-stranded nucleic acids, DNAzymes, affinity ligands, whole cells, coenzymes, enzyme-mimicking metalloproteins, enzymes, aptamers, antibodies biological receptors, and molecular imprinted polymers (MIPs) which act as artificial receptors; (2) the use of electrocatalytic electrode-coating and conducting films which improve the electrochemical signal response interference of other analytes, (3) nanostructure interfaces which reduce the signal-to-noise ratio and improve the sensitivity of a sensor, and (4) chemometrics which eradicate the signal overlapping with the interfering analytes and other small molecules. The analytical performance of the biosensor could be enhanced by implementing these approached while constructing the sensors.

Further, the biosensor technique has improved the growth of nanoscience and nanotechnology. The capability to alter and control the materials at an atomic level, i.e., nanometre scale, and consequently understand its basic process at the nanoscale led to innovative biosensor development. Nanomaterials (NMs) are generally categorized on the basis of their dimensions such as zero-dimensional (0D), first (1D), second (2D), and third (3D) nanomaterials. The dimensions of the nanomaterials play an essential role in determining their physical, chemical, biological, electrical, and optical properties [36]. With advancements in nanotechnology, research and development (R&D) in biosensors has emerged as a multidisciplinary and open field. Nanotechnology breakthroughs have accelerated the development of medical diagnostic assays and devices that are quicker, economical, highly sensitive, and much more accurate. Nanomaterial-based biosensors combine diverse fields such as chemistry, molecular engineering, material science, and biotechnology. They have such great sensitivity that specific biosensors can potentially detect as little as one parasite per microlitre of blood. Nanotechnology combined with genomics, proteomics, and molecular machinery can contribute to the creation of efficient, dependable, and quick onsite medical diagnostics. Nanotechnology has transformed these techniques by facilitating the fabrication of innovative materials for developing medical diagnostic instruments. Using nanoscience to regulate, manage, and integrate atoms and molecules to construct nano-dimensional (1–100 nm) structures and components open up new avenues for diagnostic-device development. Because of their tiny structures and high surface area to volume ratio, they have precious properties. By exploring different NMs, such as nanocomposites (dendrimers), quantum dots (QDs), nano-wires, nano rods, carbon nanotubes (CNTs), graphene dots, and nanoparticles (NPs) (metal and metal oxide-based) with distinct characteristics offers the chance to improve the performance of the biosensor and LOD by controlling the morphology and size of NMs [40,41].

### 2.1. Biosensors for Human Metabolites Detection

In metabolism, the intermediate and final products are the metabolites. These metabolites can be glucose, urea, uric acid, lactates, cholesterol, creatine, hydrogen peroxide, ketone bodies, creatinine, hypoxanthine, xanthine, etc. There have been some significant breakthroughs via electrochemical sensors in clinical applications ceoncerning one of the major constituents, glucose. The economic and sturdy nature of commercially available electrochemical glucometers is a strong indicator of their success. Here we will discuss some of the significant developments of biosensors in different metabolites as summarized in Table 1.

#### 2.1.1. Glucose

Diabetes mellitus is recognized as a global-wide health issue. This kind of metabolic disorder is due to insulin insufficiency in the blood and is related to fluctuations in blood glucose concentrations in the optimum range (4.4–6.6 mM) [42]. Initially, the commercially available glucose sensors had glucose oxidase enzyme immobilized over an electrode surface. Moderators were used at the surface of the electrode interface and enzyme for enhanced electron transport [43]. In the third generation of glucose sensors, direct electron transfer was instigated at the interface [44]. These sensors had outstanding specificity, although there were issues related to denaturation of the enzyme and reduced reproducibility after storage [45]. The non-enzymatic glucose sensors can be used as an alternative to overcome these shortcomings. However, they work in an alkaline environment unsuitable for in vivo applications.

Recently, different analytes have been utilized instead of blood for glucose monitoring, such as an enzymatic amperometric sensor that detects glucose fluctuations in tears [46]. The sensor was based on glucose oxidase immobilization over the electrode surface (iridium/platinum wire of 0.25 mm) and the release of hydrogen peroxide (H_2_O_2_) caused by the enzymatic reaction was detected at the anode, as shown in Figure 2iv. To reduce the interferent intrusion and enhance specificity in tear fluids, nafion and an electro-polymerized film of 1,3-diaminobenzene/resorcinol have been used as inner layers. The limit of detection (LOD) of the glucose sensor was 1.5 μM with a sample volume of just 4–5 μL.

For detection of glucose fluctuations in different biological fluids such as urine, serum, and blood, a microfluidic paper-based electrochemical device (μPED) was fabricated. The μPED consists of microfluidic channel patterns made through wax-printing or photolithography and screen-printed electrodes (SPE) utilizing conducting inks (such as Ag/AgCl or carbon) over the paper surface [47] [Figure 2iii]. Glucose oxidase was set over the paper microchannel. Through chronoamperometry (CA), a linear calibration was made for glucose levels up to 22.2 mM. A different type of paper-based sensor was fabricated for conducting a flow-injection glucose study [48]. In this system, the buffer solution from the elevated reservoir moves to the sink below through gravity-driven capillary transport. The nitrocellulose membrane was utilized as an intermediate for glucose oxidase immobilization over a thin platinum electrode fixed over a rigid surface. The reference and counter electrodes were set in an elevated reservoir. CA made it possible for glucose monitoring of up to 26 mM, with 0.2 mM LOD as shown in Figure 2ii. This technique is independent of throwaway test strips.

Solanki et al. have used polyvinyl alcohol capped copper oxide (PVA-CuO) NPs to fabricate novel capacitive biosensors to detect glucose concentrations non-enzymatically [49]. The designed nano biosensor showed a decline in capacitance measurements with increasing glucose concentration. Further, a highly specific capacitive sensor based on the thin film of PVA-CuO over ITO using ARDUINO UNO has been developed by Solanki et al. for non-enzymatic detection of glucose [50]. The obtained studies showed a decline in capacitance measurements with increasing glucose concentrations. The reproducibility and repeatability of the developed system were five chips and six readings, respectively, with a shelf-life of 4 weeks. Moreover, Solanki et al. fabricated novel biosensing platforms for immobilizing GOx over gelatin B [47] and agar [48] ionogels-modified ITO electrodes to detect glucose concentration. Both ionogels have been made in ionic liquid solutions such as 1-octyl-3-methyl imidazolium chloride and 1-ethyl-3-methylimidazolium chloride. The gelatin B ionogels-based bioelectrodes displayed a detection range from 1–20 mM, LOD of ≈ 0.174 mM having a sensitivity of 4.6 µA mM^−1^ cm^−2^. The agar-ionogels-modified ITO electrodes (GOx/Ag–C2/ITO and GOx/Ag–C8/ITO) showed improved sensitivity of 14.6 μA mM^−1^ cm^−2^ and ≈4.1 μA mM^−1^ cm^−2^ with a linear range of 0.28–5.6 mM and 0.27–16.7 mM, respectively.

A worldwide mobile electrochemical detector (μMED) was fabricated to utilize in different applications with few resources [51]. This device can be operated with a wide variety of electrode types which can be connected to the Internet using any smartphone [Figure 2i]. The μMED had glucose oxidase-modified test strips for directly sensing glucose in the blood. CA was utilized to measure glucose from 2.8 to 27.8 mM in a linear range. Direct integration of the sensing platform over glucose renewable sensor strips has been performed using a roller pen with enzyme-based ink [52]. This enzyme-based ink pen allowed a simplistic facile creation of excellent economic amperometric sensors with a malleable design for different surfaces. The renewable sensor strips show excellent repeatability of up to 20 cycles of 2 mM glucose with constant high sensitivity in blood.
Figure 2(**i**) Image showing μMED interfaced with a cellular device and commercial glucose test strip for data transmission over voice via an audio cable [Adapted with permission from ref. [52], Copyright 2014 National Academy of Sciences of the United States of America]; (**ii**) Schematic representation of the electrochemical detection of glucose in a paper-based flow system [Adapted with permission from ref. [48], Copyright 2012 ACS]; (**iii**) Scheme depicting the development of amperometric glucose biosensor for quantification of glucose levels in tears [Adapted with permission from ref. [47], Copyright 2011 ACS]; and (**iv**) Scheme depicting a paper-based electrochemical sensing device for glucose detection [Adapted with permission from ref. [46], Copyright 2010 RSC].
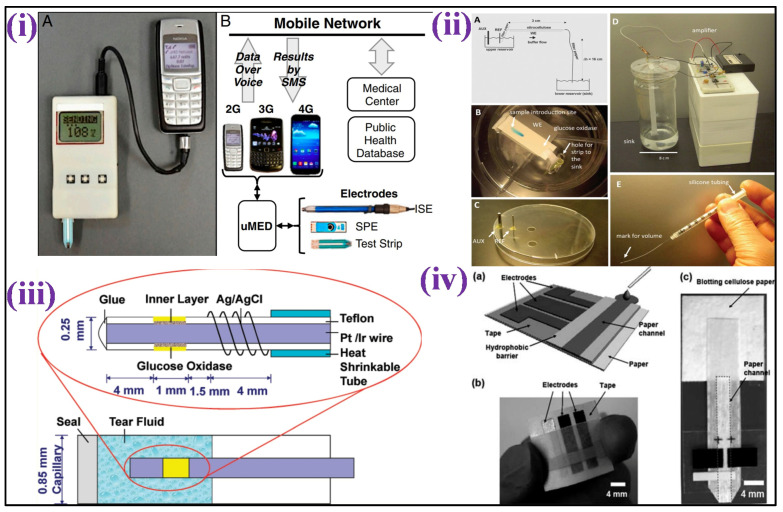


The plant lectin Concanavalin A (Con A) has been the focus of considerable research efforts to fabricate affinity-based glucose sensors because of its intrinsic activity and stability under physiological pH [47,53]. For example, gold nanoparticles (GNPs) coated with Con A were constructed on a gold electrode with polytyramine modification to create a competitive capacitive glucose sensor [54]. The biosensor showed a linear response to concentrations of glucose ranging from 1 μM–10 mM under optimum parameters according to the results. Another non-enzymatic glucose sensor was developed with localized pH modulation within a symmetric Au–Au junction electrode [55]. This study made a paired Au–Au joint electrode by electro-depositing gold onto two platinum disc electrodes kept close together. To elevate the pH locally, the working electrode’s potential was fixed to 1.5 V against the saturated calomel electrode (SCE). The sensor response was linear within the physiological glucose range of 1–10 mM, according to CV and SWV measurements.

Moreover, Bansi D Malhotra, also known as the father of Indian biosensors, has contributed significantly to the development of glucose biosensors using various types of nanomaterials. Malhotra and his co-workers proposed an enzymatic glucose sensor utilizing glucose oxidase (Gox) attached onto sol–gel derived CeO_2_-TiO_2_ nanocomposite that was electrophoretically deposited onto an ITO electrode [56]. The sensor showed a wide linearity (0.56–22.2 mM) and a sensitivity of 53.48 µA mM^−1^ cm^2^ with a LOD of 10 mg dL^−1^. Further, nano-structured polyaniline (NS-PANI) has been electrophoretically deposited over ITO to immobilize GOx to design a glucose biosensor by Malhotra et al. The developed biosensor displayed excellent sensitivity of 1.05 × 10^−4^ mA mg^−1^ dL, linear range up to 400 mg dL^−1^ having a LOD of 2.1 mM [57]. Further, Malhotra et al. developed a glucose sensor using GOx immobilized over thin films of cerium oxide (CeO_2_) NPs deposited onto platinum (Pt) coated glass plate using pulsed laser deposition (PLD) [58]. The GOx/CeO_2_/Pt biosensing platform showed a wide detection range of 25–300 mg dL^−1^ towards measuring glucose concentration with a LOD of 1.01 mM.

Recently, several nanocomposite-based biosensors have been reported for glucose detection using Cu/Cu_2_O/NiO hybrid for serums and beverages [59], wearable glucose determination in interstitial fluid of rats [60], and MXenes (Ti_3_C_2_T_x_)-based nanocomposites with Cu_2_O [61].

#### 2.1.2. Lactose and Galactose

Lactose is a disaccharide broken down into monosaccharides, glucose, and galactose, via the lactase enzyme within the gastric tract. Lactose levels in the blood can be used to determine whether or not a patient has gastrointestinal cancer [62].

An enzyme-based lactose amperometric sensor was fabricated by immobilization of β-galactosidase enzyme in the stratified film of poly(vinylsulfonate) and poly(ethyleneimine), as shown in Figure 3i. The film was deposited over a Prussian blue-covered ITO electrode [63]. This sensor led to a linear detection for lactose concentrations and had 1.1 mM as LOD. The other lactose-specific enzyme, i.e., cellobiose dehydrogenase, was used to develop an amperometric-based lactose sensor [64]. The enzyme was made to attach with MWCNTs and poly(3-amino-4-methoxybenzoic acid-co-aniline) nanohybrid followed by its deposition on a gold electrode surface. The sensor achieved linearity for lactose concentrations up to 30 mM [Figure 3ii]. Further, a lactose biosensor based on Langmuir–Blodgett (LB) films of poly(3-hexyl thiophene) (P3HT)/stearic acid (SA) has been fabricated by Malhotra et al. by immobilizing lactase and galactose oxidase (GaO) for quantification of lactose in milk and its products. The bioelectrodes showed linearity of 1–6 g dL^−1^ with a shelf life of more than 120 days [65].

A galactose amperometric biosensor was developed by Malhotra et al. using mono-enzyme GaO for the detection of galactose in milk and milk products [66]. The P3HT/SA was used to immobilize GaO on an ITO-covered glass substrate utilizing Langmuir–Blodgett (LB) film deposition approach. The fabricated enzyme biosensor showed a linear detection range of 1–4 g dL^−1^ galactose. In another work, Malhotra et al. proposed an enzyme biosensor for the determination of galactose in human serum by Malhotra et al. [67]. Monolayers of poly(3-hexyl thiophene) were formed by dispensing a mixed solution of stearic acid in chloroform. The fabricated galactose biosensor presented a linear response in blood serum from 0.05–0.5 g galactose L^−1^. Moreover, some recently developed biosensors are also reported for electrochemical detection of lactose [68,69].

#### 2.1.3. Uric Acid (UA)s

Uric acid, i.e., UA, is the most abundant nitrogenous element present in urine that is linked to several medical conditions. Other pathological and gout problems such as kidney, diabetes, heart disease, high blood pressure, and obesity may be caused by an increased UA concentration in the blood, commonly known as Lesch–Nyhan syndrome or hyperuricemia. UA concentrations in the blood should be between 200 and 400 μM [70]. The intrusion generated by other biomolecules with similar oxidation potentials, such as ascorbic acid, is one of UA’s most significant issues with direct biological measurement [71].

An amperometric enzymatic-based biosensor for UA was developed with a glassy carbon electrode (GCE) with Nafion film and ferrocene-induced electro-activated uricase enzyme deposited inside [72]. Uricase electro-activation was attained by the CV arising from the electrostatic interaction between the tryptophan and Fc molecules that remain inside the enzyme’s hydrophobic pockets. This biosensor had a linearity (500 nM to 600 μM) with a 230 nM detection limit. Using boron-doped diamond (BDD) microelectrodes, a non-enzymatic test for UA in urine was devised, as shown in Figure 3iii [73]. BDD’s outstanding features, such as low non-specific adsorption of other species, minimal background current, antifouling characteristics, and excellent stability after electrochemical reactivation, are only a few advantages used to support the strategy. When ascorbic acid is present, its direct determination of UA values down to 1 mM was possible. In a similar method, a gold electrode coated with sulfur-adlayer was fabricated for UA detection. The sensor aided oxidation of UA at a considerably low overpotential with an increased current density with reference to an unmodified electrode [74]. This electrochemical sensor showed linearity for UA between 2.5 μM to 5 mM, having a detection limit of 0.4 μM. A sensor modification with dendrimer was fabricated for specific detection of UA when ascorbic acid is present [75]. In this sensor, a GCE was consecutively changed with poly(amidoamine) (PAMAM) dendrimers containing 128-COOH exterior groups and cysteamine-capped GNPs. CV was used for the detection of UA, with 34.5 nM LOD. Further, Malhotra et al. fabricated an amperometric biosensor after immobilization of uricase using the excellent properties of carbon nanotubes (CNTs) to detect UA concentration in serum samples. The developed enzymatic biosensor showed a rapid response of 8 s and LOD of 5 μM, having the stability of >180 days due to the employment of CNTs [76].

Recently, a non-enzymatic platform for electrochemical detection of UA was developed by Santos et al. in human urine [77]. In addition, Harraz and his co-worker’s reported a novel polypyrrole-carbon black-Co_3_O_4_ (PPy-CB-Co_3_O_4_) nanocomposite for a non-enzymatic UA sensor using a modified glassy carbon electrode which showed linearity ranging from 0.75–305 μM having sensitivity (0.8786 μA μM^−1^ cm^−2^), and lowest LOD of nearly ~ 0.46 μM in human blood serum [78].
Figure 3(**i**) The biosensor fabricated with a (PEI/β-Gal)n LbL film for lactose detection operating at 0.0 V vs. SCE [Adapted with permission from ref. [63], Copyright 2014 ACS]; (**ii**) Au-MPA-[MWCNT-P(AMB-A)-PDA]/CDH electrode showing amperometric current response with the concentration of lactose [Adapted with permission from ref. [64], Copyright 2014 MDPI]; and (**iii**) proposed schemes for oxidation of UA in the absence/presence of AA in which (I) is UA, (II) bis-imine compound, (III) iminealcohol compound, and (IV) uric acid-4,5-diol [Adapted with permission from ref. [73], Copyright 2012 ACS].
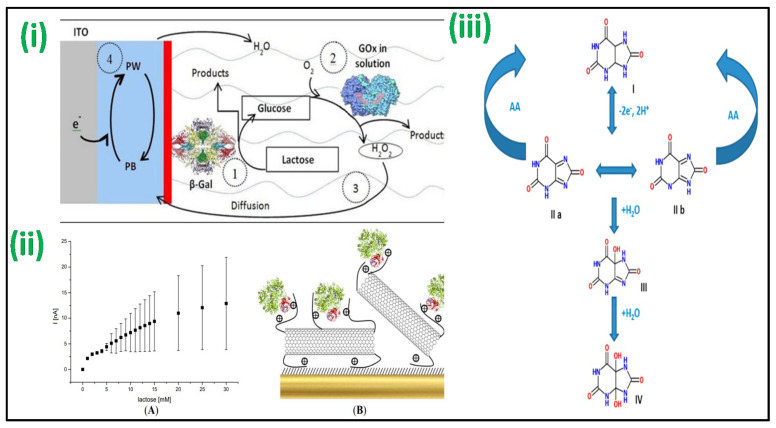


#### 2.1.4. Urea

Urea results from protein metabolism, so its concentration can reveal a lot about a person’s nutritional state. Urea concentrations should be between 2.5 and 6.6 mM, based on the person’s age and gender. The urine urea nitrogen (UUN) and blood urea nitrogen (BUN) tests are frequently utilized to diagnose kidney problems such as end-stage and acute renal kidney failure. Yet, a change in urea concentration is not always due to kidney disease; it might be due to dehydration or increased protein intake [79].

Several works have been reported recently for electrochemical detection of urea [80,81]. Further, both urea and glucose can be measured utilizing a pH-switchable electrochemical sensor [82]. The detection is based on in situ pH-switchable enzyme-catalyzed processes in which urease catalyzed urea hydroxylation or glucose oxidase catalyzed glucose oxidation caused a change in the electrolyte’s pH, resulting in a distinct electrochemical response toward redox couple. Urea analysis was performed with DPV, CV, and EIS for the linear concentration (1 to 7 mM). In an enzyme-based method, an amperometric urea sensor was developed with an ITO electrode coated with multi-layered graphene [83]. Multi-layered graphene exhibited enhanced 2D surface area with improved electrical conductivity. Immobilization of urease and glutamate dehydrogenase enzymes was performed over the sensor’s surface that showed linearity for urea from 1.7 to 16.7 mM, with a 0.6 mM detection limit. In another study, a thin film of gelatin organogel-based nanocomposite (GA-NC) was made onto an ITO (GA-NC/ITO) electrode by drop-casting and was utilized to detect urea. The developed GA-NC/ITO electrode exhibited a sensitivity of 5.56 and 32.7 µA mM^−2^ cm^−2^ in the two-concentration range of 2 to 20 and 0.1 to 2 mM, respectively [84]. A potentiometric urea sensor based on urease has also been reported [85]. Immobilization of the enzyme was performed by physisorption over the electrodeposited polyaniline film or by attaching to the layered film of alternating chitosan and carboxymethyl pullulan made over a polyaniline film. The current response was linearly proportional to concentrations of urea ranging from 1 mM to 100 mM. Based on the enzyme implanted layered film, the sensor has better stability, which can be attributed to the polysaccharide coating’s protective quality. A gold electrode covered with chitosan-doped cadmium sulphide QDs was used as the primary substrate in an MIP-based urea voltammetric sensor [86]. Two linear ranges between 5 pM and 0.4 nM and 0.5 and 70 nM, for urea determination were shown using DPV, with 1 pM LOD.

Moreover, Bansi D Malhotra designed and developed various nanocomposites-based enzymatic bioelectrodes for urea detection. For instance, titania–zirconia (TiO_2_–ZrO_2_) nanocomposite integrated with a microfluidics system has been utilized for co-immobilization of urease (Urs) and glutamate dehydrogenase (GLDH) for the detection of urea [87]. The bioelectrode Urs-GLDH/TiO_2_-ZrO_2_/ITO showed a sensitivity of 2.74 μA [log mM]^−1^ cm^−2^ and a broad linearity of 5–100 mg dL^−1^ with LOD of 0.44 mM. In another work, mesoporous silica particle anchored with graphene oxide (GO) hybrid system was used for urea detection having a sensitivity of 2.6 μA mM^−1^ cm^−2^ with LOD of 14 mg dL^−1^ [88]. Additionally, glutamate dehydrogenase (GLDH) and urease (Urs) were covalently co-immobilized onto a self-assembled monolayer (SAM) comprising 10-carboxy-1-decanthiol (CDT) immobilized onto patterned gold (Au) electrodes for determination of urea utilizing poly(dimethylsiloxane)-based (PDMS) microfluidic channels. The sensor showed a linearity ranging from 10–100 mg dL^−1^, with LOD of 9 mg dL^−1^ and high sensitivity of 7.5 μA mM^−1^ cm^−2^ [89]. Furthermore, an electrochemically immobilized film of biocompatible zirconia (ZrO_2_) on gold-covered glass electrodes was used to make a urea biosensor by immobilizing Urs and GLDH. The urea biosensor achieved a linear response of up to 40 mg dL^−1^ and sensitivity of 0.071 μA/(mM cm^−2^) having stability of 4 months when kept in 4 °C [90].

#### 2.1.5. Cholesterol

Cholesterol monitoring is critical, especially for patients with a higher threat of heart disease. Blood cholesterol levels must be measured to diagnose atherosclerosis and estimate the threat of thrombosis and cardiovascular disease. Total cholesterol levels in the blood should not exceed 5.17 mM; however, this value varies with age, weight, and gender. A measurement of 5.17–6.18 mM should be the threshold amount for total cholesterol concentration in a patient’s blood, which is regarded as very high [91]. Bansi D Malhotra made an enormous contribution towards the development of cholesterol biosensors. He developed several cholesterol biosensors via immobilization of cholesterol oxidase (ChOx) enzyme onto ITO electrodes modified with different types of nanomaterials and nanocomposites [92,93,94,95,96,97,98,99,100].

Mixing MIP-coated MWCNTs, silicon alkoxide and graphite powder, then collecting the resulting product as the electrode opening of a Teflon sleeve, a voltammetric cholesterol sensor-based MIP was constructed [101]. Linear sweep voltammetry (LSV) was utilized for cholesterol measurement depicting linearity from 10 to 300 nM, with a 1 nM detection limit. A glutaraldehyde-functionalized magnetic ferrite nanoparticles (GA-MNPs) and poly-(diallyldimethylammonium chloride)-coated MWCNTs (PDDA/MWCNTs) were used to fabricate an enzymatic amperometric cholesterol sensor [102]. Before adding GA-MNPs, the carboxylated MWCNTs were wrapped with the +ve charged PDDA. On the composite cast, the enzymes ChOx and HRP were co-immobilized and deposited over a GCE. The sensor showed a calibration curve range (0.01 to 0.95 mM) with a 0.85 μM detection limit. The same substrate was used for powering the cathodic and anodic electrocatalytic reactions, resulting in a reagent-free self-powered cholesterol detection platform [103] [Figure 4ii]. Immobilization of cholesterol oxidase was performed in a sol–gel matrix, which was used to coat both the anode and cathode. At the anode, cholesterol oxidation catalysation is performed by cholesterol oxidase, whereas, at the cathode, Prussian blue electrocatalytically decreased H_2_O_2_ due to the conversion of cholesterol enzymatic. The self-driven amperometric sensor was capable of measuring concentrations of cholesterol up to 4.1 mM.

A sensor based on amperometric triglyceride was made by mixing four enzymes that include glycerol-3-phosphate oxidase, HRP, lipase, and glycerol kinase, co-immobilized over the platinum electrode surface with poly(vinyl alcohol) membrane, a coating in another enzymatic technique [104]. In the range of 560 μM to 2.3 mM, the current response was proportional to the triglyceride concentration, with 210 μM LOD. A facile electrochemical micro paper-based analytical device (EμPADs) and a commercial glucometer were integrated innovatively for cholesterol determination [105]. The sample and reference zones were defined by printed wax barriers on each EμPAD, while the experimental strips were constructed of a single piece of paper or plastic. ChOx catalyzed the cholesterol to cholest-4-en-3-one conversion in the enzyme-mediated stage with simultaneous reduction of Fe(III) to [Fe(II) (CN)_6_]^4−^ ions, and the Fe(II) formed was electrochemically evaluated. An amperometric cholesterol study showed a linear plot of 0.5 to 5.2 mM in human plasma with 340 μM LOD. A non-enzymatic voltammetric cholesterol sensor platform has also been published [106]. An electroactive and hydrophilic cholesterol detecting interface was created by altering a chemically transformed graphene with β-cyclodextrin and methylene blue. Methylene blue was eradicated from the matrix and transferred to buffer solution in the presence of cholesterol, where it was electrochemically detected. A linear curve ranging from 5–100 μM was developed using DPV, with 1 μM LOD as shown in Figure 4i.

#### 2.1.6. Lactate

In the clinical investigation of acute cardiac disorders, respiratory insufficiency, and shock, the lactate level in the blood was used. lactic acidosis is also known to occur in the presence of medication toxicity, left ventricular failure, and tissue hypoxia. Lactate physiological levels in the blood are between 0.5 and 2.2 mM [107].

Recently, flexible laser scribed graphitic carbon [108] and dual functional electrochemical sensor using macroscopic film of polyaniline-based (PANI) biosensors were reported for lactate detection [109]. The enzyme named Lactate oxidase was immobilized in the matrix of albumin/mucin hydrogel prepared over a platinum electrode surface to create an amperometric lactate biosensor [110]. By wrapping the electrode with a membrane of Nafion, the intrusion due to anionic species was reduced. In the region of 2 to 1000 μM, linearity of catalytic current based on lactate concentration was validated, with 0.8 μM LOD. The NAD^+^ and lactate dehydrogenase enzyme were co-immobilized over the GCE surface pre-coated with Fe_3_O_4_ magnetic nanoparticles and MWCNTs nanocomposite in another enzymatic technique. Lactate showed linearity ranging from 50 to 500 μM using a DPV technique, with 5 μM LOD [111]. For non-invasive lactate real-time monitoring, a printed amperometric sensor “tattoo” for the wearer’s skin was fabricated in sweat while exercising [112] [Figure 4iv]. The wearable sensor based on lactate oxidase displayed specificity for lactate showing a linear range to 20 mM and robustness towards epidermal wear. The gadget was tested on humans, and it was discovered that the lactate profiles accurately mirrored lactate generation in sweat at various exercise intensities. Using mixed *Streptococcus thermophilus* and *Lactobacillus bulgaricus* bacterial cultures and palygorskite, a bacterium stabilizing matrix attached over the O_2_ electrode surface, whole cells were employed as a sensing probe for monitoring lactate [113]. The sensor’s amperometric response was linear, ranging from 0 to 300 μM. Two components were used to create an FET-based lactate sensor: (i) a pH-sensitive chemical field-effect transistor (FET) and (ii) a metallic microelectrode kept on sensitive gate [114], as shown in Figure 4iii. At the microscale, the presence of these two components is coupled with potentiometric and amperometric effects. Lactate may show linearity from 1 to 6 mM using this sensor.

The electrochemical entrapment of polyaniline (PANI) onto sol–gel-derived tetraethylorthosilicate (TEOS) films deposited onto ITO-coated glass by Malhotra et al. has been utilized for immobilization of lactate dehydrogenase (LDH) [115]. The amperometric response of the electrodes under optimum conditions exhibited a linear relationship from 1 mM to 4 mM. In another study for the immobilization of LDH for lactate detection, TEOS-derived sol–gel films were used. It was observed that TEOS-derived sol–gel films with physisorbed LDH had a linear response in the range of 0.5–4 mM, but those with LDH in a sandwich configuration had a linear response ranging from 0.5–3 mM l-lactate [116]. Further, the immobilization of LDH on electrochemically polymerized polypyrrole–polyvinylsulphonate (PPY–PVS) films was achieved to develop a lactate biosensor by Malhotra et al. [117]. The fabricated PPY–PVS–LDH electrodes showed LOD of 1 × 10^−4^ M, a linear range of 0.5 to 6 mM for L-lactate detection, a response time of about 40 s, and stability of 2 weeks. Lactate oxidase (LoD) and lactate dehydrogenase (LDH) have been co-immobilized on electrochemically prepared PANI toward the detection of lactate by Malhotra et al. [118]. The PANI/LOD/LDH electrodes showed a linearity from 0.1–1 mM, having a detection limit of 5 × 10^−5^ M, and stability of about 3 weeks.
Figure 4(**i**) It demonstrates the cholesterol sensing mechanism, utilizing Grp-β-CD as the working substrate [Adapted with permission from ref. [106], Copyright 2015 Elsevier]; (**ii**) scheme showing a self−powered biosensor for measurements of cholesterol [Adapted with permission from ref. [103], Copyright 2014 ACS]; (**iii**) cross section (a)and details (b) of the ElecFET device for lactate detection (chip size: 3.5 × 3.5 mm^2^) [Adapted with permission from ref. [114], Copyright 2013 Elsevier], and (**iv**) illustration of tattoo-based electrochemical biosensors for non-invasive lactate detection in human samples [Adapted with permission from ref. [112], Copyright 2013 ACS].
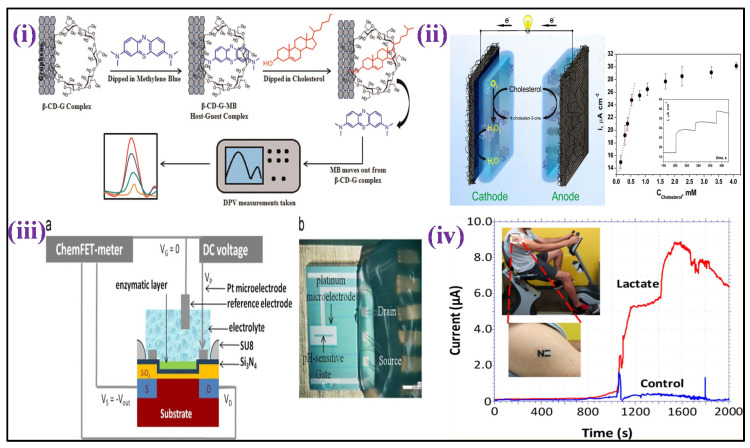


#### 2.1.7. Hydrogen Peroxide (HP)

Reactive oxygen species, i.e., ROS, are considered intracellular signaling molecules that regulate cell apoptosis and DNA damage, among other things. On the other hand, the ROS accumulation in cells causes oxidative stress, which has been related to neurological illnesses, cancer, autoimmune diseases, and Alzheimer’s disease. Hydrogen peroxide (HP) is a commonly researched type of ROS in cells because it causes a batch of harmful biological changes. To completely understand the reactive species roles of cell physiology, specific HP detection in cells and evaluation of its release mechanism are required [119]. Malhotra et al. [120,121,122] and others [123,124] fabricated a few enzymatic biosensing platforms for the detection of HP via immobilization of horseradish peroxidase enzyme.

An enzymatic voltammetric HP biosensor was created by HRP enzyme insertion into a matrix of cross-linked 3D polymer carrying a moveable osmium redox (Os^+/+2^) and allowing substrate diffusion. Collisions of the redox mediators delivered [125] electrons step-by-step to the enzyme from the electrode surface, reducing the analyte. According to CV, HP has a dynamic linearity ranging from 1–100 μM. Furthermore, HP detection via CA showed linearity from 1–10 nM, having a 1 nM LOD. Mesoporous carbon nanoparticles were employed to construct electrically linked enzyme electrodes to sense HP in a similar method [126]. Methylene blue was used to fill carbon NPs pores, which were subsequently HRP capped and fixed on the surface of GCE detecting up to 25 mM HP using CV.

A microperoxidase-11 enzyme mounted on a top MWCNTs-functionalized bacterial cellulose was used to create another enzymatic amperometric HP sensor [127]. A dynamic conductive film having great biocompatibility was created by modifying bacterial cellulose with MWCNTs. The proposed sensor could measure HP levels throughout a linear detection range (0.1–257.6 μM), having a detection limit of 0.1 μM. A catalase enzyme mounted on a top film of MWCNTs and L-lysine -modified GCE [128] was used to create an enzymatic amperometric sensor [Figure 5i]. The concentration from 1 μM–3.6 mM showed a linear current response to HP, with 8 nM LOD.

Because of the innate metal center activity inside the enzyme molecule, metalloproteins are commonly used as an HP sensing element. For example, a protein microarray was created using L-cysteine-modified Au-TiO_2_ hydrophilic micropatterns, followed by immobilization of metalloprotein Cytochrome c [129]. Wide linearity for HP ranging from 1 nM to 10 mM was measured using amperometry, having a LOD of 2 nM, as shown in Figure 5ii. Using an imidazole-terminated self-assembled monolayer, HP amperometric sensor was recently constructed by directly connecting the heme center of Cytochrome c with the surface of gold ultra-microelectrode [130]. The HP level can be determined using this sensor with 3 μM LOD.

Another metalloprotein, hemoglobin, was used to fabricate an HP amperometric sensor via hemoglobin microbelts electrospinning over the surface of GCE [131]. Then, hemoglobin catalyzed the HP reduction over a linear range (10–230 μM) having a detection limit of 0.6 μM. Myoglobin was immobilized on an ITO covered with nanostructured porous cerium dioxide, a layer in another metalloprotein-based HP sensor [132]. With linearity of 3 mM and a LOD of 0.6 M, this sensor could detect HP. A heme peptide deposited onto a conductive 3D hybrid film of CNTs on graphene surface was also used to create an amperometric sensor for HP [133]. The film’s microvoids helped HP diffuse to the altered heme peptide, which resulted in a higher cathodic current. Over a range of 1–30 μM, the cathodic current increases in linear proportion with the HP concentrations. A derivative of zinc porphyrin-fullerene (C60) was encapsulated in a tetraoctyl ammonium bromide film and stored onto the surface of GCE [134] to provide another amperometric HP sensor. HP electrocatalytic reduction had wide linearity (35 μM–3.4 mM), having a LOD of 810 nM. Nanomaterials’ electrocatalytic activity has also been used to produce an HP sensing platform. By hydride reduction of manganese acetylacetonate in the presence of GNPs, core/shell Au/MnO nanoparticles were synthesized, which were then used to develop an amperometric HP sensor [135]. The HP was linearly determined ranging from 20 nM to 15.1 mM, having LOD of 8 nM.

An electrochemical HP sensor was created utilizing nitrogen-doped carbon nanotubes in a similar method (NCNTs) [136]. Compared to NCNTs and MWCNTs, it showed a robust electrocatalytic activity via a four-electron transfer procedure for reduction of O_2_, while MWCNTs-mediated electrocatalysis uses a two-electron transfer process. This amperometric sensor offers a linear HP-level detection range (1.8 to 139 μM), having 370 nM detection limit. A 3D graphene network, i.e., 3DGN, was synthesized using chemical vapor deposition (CVD) in another HP nanomaterial-based experiment [137]. Pt nanoparticles/3DGN, MWCNTs/3DGN, Pt/MWCNTs/3DGN, and MnO_2_/3DGN were among the composites created using the 3DGN as a template. The Pt/MWCNTs/3DGN-based amperometric sensor showed a linear detection range for HP from 25 nM to 6.3 μM, with 8.6 nM LOD.

#### 2.1.8. Ketone Bodies

Ketone bodies include acetoacetate, acetone, and 3-hydroxybutyrate. These biomarkers are related to diabetes, which points to critical ketoacidosis.

Lee and his collaborators have recently reported an enzymatic biosensor using gold (Au) electrode acetylacetic acid (AcAc) determination in a urine sample [138]. In addition, Yadav et al. has developed a perovskite BaSnO_3_-based electrochemical biosensor for acetone sensing in the human breath [139]. Further, an enzyme-based voltammetric sensor for detection of 3-hydroxybutyrate was fabricated with immobilization of the enzyme (3-hydroxybutyrate dehydrogenase) over the SWCNTs-covered surface of SPE [140], as shown in Figure 5iii. CV was employed to analyze 3-hydroxybutyrate by detecting the signal from the production of 3-nicotinamide adenine dinucleotide (NADH) (enzymatic reaction product) by performing CV. A linear response was presented by calculatingg electrochemical readings with an increase in 3-hydroxybutyrate concentrations ranging between 0.1 to 2 mM, having a detection limit of 9 μM. A non-enzymatic voltammetric acetone sensor was prepared with GCE coated with ZnO nanoparticles [141]. There was a linear current increase with increasing acetone concentration (130 μM–1330 mM) and a 68 μM LOD.

#### 2.1.9. Xanthine

Xanthine is considered an intermediary product of the purine nucleotide and deoxynucleotide metabolism. It is uric acid’s metabolic precursor, its initial sign of an unusual purine profile, and can hence be used as an indicator for various disorders, including hyperuricemia, gout, cerebral ischemia, and xanthinuria [142]. In urine, the physiological range of xanthine is 40 to 160 μM [143].

A xanthine enzymatic amperometric sensor was fabricated with modification of a 3D network of electro-polymerized GNPs capped with SWCNTs, p-aminothiphenol, 2-mercaptoethanesulfonic acid, and 1-adamantanethiol over GCE [144] [Figure 5iv]. β-cyclodextrin and xanthine oxidase were immobilized via host–guest interactions over the hybrid. This sensor had a linear detection of xanthine from 50 nM to 9.5 μM, having a 40 nM LOD. A non-enzymatic sensor for xanthine was fabricated with GCE altered with reduced graphene oxide (RGO) with a coating of poly(o-aminophenol-co-pyrogallol) (PAP) electropolymerized film [145]. PAP synthesis was performed through the electrochemical copolymerization of pyrogallol and o-aminophenol in an acidic medium. The concentration of electropolymerized PAP was enhanced in the presence of RGO due to its huge availability of surface area. The amperometric analysis had a linear detection range for xanthine ranging from 1 to 120 μM, having a detection limit of 0.5 μM. Recently, development of an electrochemical biosensor using vanadium tetrasulfide nanospheres (VS_4_) fixed into nitrogen-doped biomass-derived porous carbon materials (VS_4_@N-BPC) was reported for electrochemical xanthine determination [146].
Figure 5(**i**) Scheme showing the CAT/PLL/f−MWCNTs/GCE preparation [Adapted with permission from ref. [128], Copyright 2014 Elsevier]; (**ii**) schematic representation of the surface modification for preparation of protein microarrays and for guided cell adhesion [Adapted with permission from ref. [129], Copyright 2010 ACS]; (**iii**) scheme depicting the biosensor development comprising carbon nanotube-modified screen-printed electrode for HB [Adapted with permission from ref. [140], Copyright 2013 Wiley]; (**iv**) illustration of fabrication of electrochemical biosensor based on functionalized hybrid gold nanoparticles/SWCNTs for xanthine determination [Adapted with permission from ref. [144], Copyright 2012 ACS].
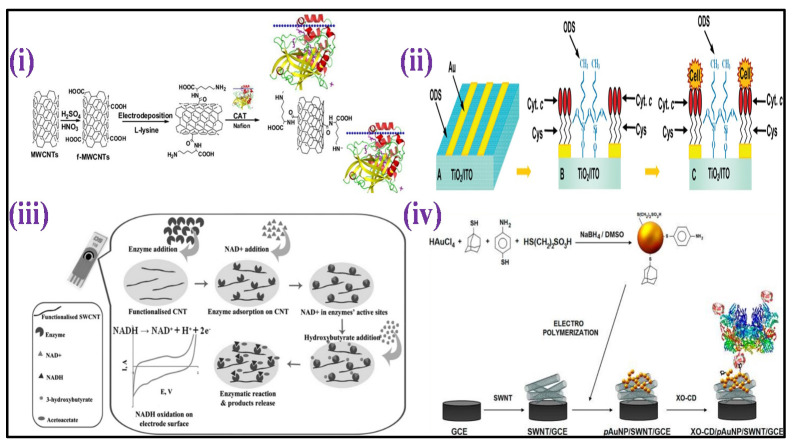


#### 2.1.10. Hypoxanthine

Xanthine is a hypoxanthine product whose physiological level ranges from 10 μM to 30 μM in blood [147]. Hypoxanthine, via the production of oxygen radicals, plays an important part in post-hypoxic reoxygenation cell injury by and works as a hypoxia indicator [148].

An enzyme-based low-potential amperometric sensor for hypoxanthine and xanthine was fabricated with xanthine dehydrogenase enzyme immobilization on a PG electrode which is an edge plane [149]. The enzyme oxidizes the hypoxanthine to xanthine, followed by xanthine conversion to UA d through oxidative hydroxylation process. The electrode exhibited a proportional behavior to xanthine and hypoxanthine concentration, showing linearity from 10 μM to 1.8 mM, having a 0.25 nM detection limit.

#### 2.1.11. Creatine

Creatine is a derivative of amino acid produced in the liver, kidneys, and pancreas from S-adenosylmethionine, L-arginine, and glycine. The stored creatine in muscular tissues is mainly transformed to phosphocreatine during the reversible reaction involving adenosine triphosphate (ATP), catalyzed via creatine kinase enzyme. Phosphocreatine is the body’s primary energy storage type. Creatinine levels in the blood and urine are utilized as a clinical diagnostic of muscle injury [150]. A combination of sarcosine oxidase and creatinase were co-immobilized on a carbon paste electrode (CPE) surface altered with Fe_3_O_4_ nanoparticles and used to make an enzymatic amperometric sensor for detection of creatine [151]. The sensor’s linearity ranges from 0.2–3.8 μM and 9 to 1200 μM, respectively, having a 0.2 μM detection limit.

#### 2.1.12. Creatinine

Creatinine is a waste product of muscle metabolism used to assess renal function. A biochemical mechanism including creatine, phosphocreatine, and ATP produces creatinine. Creatinine levels should range from 53–115 μM in the blood.

By co-immobilizing the three enzymes, an enzymatic amperometric sensor was developed for creatinine: sarcosine oxidase, creatinine aminohydrolase, and creatine aminohydrolas on ZnO nanoparticles/chitosan/carboxylated MWCNTs/polyaniline coated composite film over the surface of a platinum electrode [152] [Figure 6ii]. The sensor displayed a linear response in the 10–650 μM, with a 0.5 μM LOD. A voltammetric creatinine sensor based on MIP has also been published [153]. In this study, the template molecule, i.e., creatinine was self-assembled over the Fe_3_O_4_ polyaniline NPs surface in the functional monomer aniline presence, as shown in Figure 6iii. The orderly structured MIP was formed through electro-polymerization on the surface of the electrode after magnetically directed preassembly GCE. With creatinine concentrations ranging from 20 nm to 1 μM, the DPV response was linearly detected, with 350 pM LOD.

Electro-polymerization was used to coat gold an electrode array having creatinine combined with pyrrole, resulting in a MIP-based amperometric sensor for detection of creatinine [154] [Figure 6i]. After that, the creatinine sample was combined with an anti-creatinine antibody that had been HRP labeled, and the product was applied to the electrode surface to give a viable response with the creatinine fixed on the surface of the electrode. Amperometric tests were performed in the H_2_O_2_ and 3,3′,5,5′-tertamethylbenzidine (TMB) presence. The level of creatinine was estimated up to 1 mM using a circular, square wave. A non-enzymatic electrochemical method for detecting creatinine based on picrate anions consumption measurement due to reaction among the creatinine molecules, utilized an edge plane of pyrolitic graphite (PG) electrode [155]. The current examination showed two linear dynamic detection ranging from of 0 to 6 and 6 to 11 mM for creatinine detection, with LOD of 720 μM.

Apart from these, recently Cui and his co-workers have developed a printed biosensor which is disposable for creatinine clinical determination during renal functioning [156]. In addition, a biosensor for detection of creatinine was also reported by a research group utilizing an Au electrode with graphene quantum dots suspended in Cu^2+^ ions (Such Au/Nafion-GQDs-Cu) [157]. Further, Fatibello-Filho et al. have proposed a screen-printed microcell for creatinine electrochemical detection [158].
biosensors-12-00733-t001_Table 1Table 1Nanomaterial-based biosensing platform towards clinically relevant human metabolite detection.MetaboliteNanomaterial Based Biosensing PlatformElectrochemical TechniqueLimit of Detection (LOD)Linear Detection Range (LDR)Real SampleRef.GlucoseTeflon-coated Pt/Ir wireAmperometry1.5 μM5–800 μMTear[46]µPEDsChronoamperometric1.0 ppb0.2–22.6 mMUrine[47]PVA-CuO/ITOCapacitance-0.5–20 mM-[49]uMEDChronoamperometry-50–500 mg dL^−1^Blood[51]Con A/AuCapacitance1.0 × 10^−6^ M1.0 × 10^−6^ to 1.0 × 10^−2^ M-[54]GOx/CeO_2_–TiO_2_/ITODPV10 mg dL^−1^0.56–22.2 mM-[56]GOx/NS-PANI/ITODPV2.1 mMup to 400 mg dL^−1^-[57]GOx/CeO_2_/Pt bio-DPV1.01 mM25–300 mg dL^−1^-[58]ITO/PB/(PEI/PVS)1(PEI/β-Gal)_30_Amperometry1.13 mmol L^−1^1.13 mmol L^−1^Milk [63]Au-MPA-[MWCNT-P(AMB-A)-PDA]/CDHAmperometry-0–30 mM-[64]P3HT/SA/b-gal/GaOAmperometry-1–6 gdL^−1^
[65]GalactoseP3HT/SA/ITOAmperometry-1–4 gdL^−1^Milk[66]ITOP3HT/SA/GaOAmperometry-0.05–0.5 g galactoseL^−1^Blood[67]Uric acidNaf/UOx/Fc/GCECV and DPV230 nM500 nM to 600 μMBlood[72]BDDCV1 mM250–1250 μMUrine[73]S-Au electrodeAmperometry0.4 μM2.5 μM to 5 mMurine[74]GCE/AuNp@cysteamine/PAMAMCV34.5 nM-Blood[75]Uricase/MWCNT/PANI/ITOCV5 μM0.005–0.6 mMSerum[76]UreaCS-rGO/Con AEIS
1–7 mMSerum[82]Urs-GLDH/MLG/ITOCV0.6 mM1.7–16.7 mM-[83]GA-NC/ITOCV-2–20 and 0.1–2 mM-[84]CdS QDs–MIPs/AuDPV1.0 × 10^−12^ M5.0 × 10^−12^–7.0 × 10^−8^ MSerum[86]Urs-GLDH/TiO_2_-ZrO_2_/ITOCV0.44 mM5–100 mg dL^−1^-[87]Urs-GLDH/GOS/ITOCV2.1 mM3.3–19.9 mM-[88]Urs-GLDH/CDT/AuCV9 mg dL^−1^10–100 mg dL^−1^-[89]Urs-GLDH/ZrO_2_/AuCV5 mg dL^−1^5–100 mg dL^−1^-[90]CholesterolChEt–ChOx/PPY–MWCNT/PTS/ITODPV0.04 mML^−1^4 × 10^−4^–6.5 × 10^−3^ ML^−1^Serum[93]ChEt-ChOx/4-ATP/AuCV1.34 mM and 1.06 mM25–400 mg dL^−1^-[94]ChOx/nan-NiO-CHIT/ITOCV43.4 mg dL^−1^10–400 mg dL^−1^-[95]ChOx/Glu/PANI-NT/ITOLSV1.18 mM25–500 mg dL^−1^-[96]ChOx/PANI-CMC/ITO
1.31 mM0.5–22 mM-[97]LactatePt electrodeAmperometry0.8 μM2 to ~1000 μMWhole blood[110]Fe_3_O_4_/MWCNT/LDH/NAD^+/GC^DPV5 μM50–500 μMSerum[111]ElecFETAmperometry-1–6 mM-[114]sol–gel/PANI/LDHAmperometry-1–4 mM-[115]PPY–PVS–LDHAmperometry1 × 10^−4^ M0.5–6 mM-[117]Hydrogen peroxideHRP-PANI-ClO_4_^−^/ITOCV1.984 mM3–136 mM-[120]HRP/PANI-CeO_2_/ITOCV50 mM50–500 mM-[121]HRP/NanoCeO_2_/ITOCV0.5 μM1.0–170 μM-[122]MP-11/MWCNTs–BCCV0.1 µM0.1–257.6 µM-[127]Au–TiO_2_/CysCV2 nM10^−9^–10^−2^ M-[129]Mb/CeO_2_/ITOCV0.6 μM0.2–5 mM-[132]Ketone bodiesSWCNT-modified SPCECV80 μM0.1–2 mMSerum[140]ZnO NPsI–V68 μM130 μM–1330 mM-[141]XanthineXO-CD/pAuNP/SWNT/GCECV40 nM50 nM–9.5 μM-[144]PAP/RGO/GCLSV0.5 μM1.0–120 μM-[145]EPG/XDHCV2.5 × 10^−10^ M1.0 ×10^−5^–1.8 × 10^−3^ M-[149]CreatinineZnO-NPs/CHIT/c-MWCNT/PANI/PtEIS0.5 μM10–650 μMBlood[152]Fe_3_O_4_@PANI NPsDPV0.35 nmol L^−1^2.0×10^8^–1.0× 10^6^ mol L^−1^
Urine and plasma[153]Conducting polymerAmperometric0.46 mg dL^−1^0–11.33 mg dL^−1^Blood[154]EPPGCV0.27 mM7.5–11.5 mMUrine[155]CreatineFe_3_O_4_-CPEEEIS2.0 × 10^−7^ mol L^−1^2.0×10^−7^ to 3.8×10^−6^ mol L^−1^Commercial creatine powder[151]


### 2.2. Biosensors for Gut Metabolites Detection

The human gut microbiota is composed of many coexisting species of bacteria, eukaryotic viruses, archaea, bacteriophages, and fungi that work together to form a diverse ecosystem that can impact human physiology, behavior, and pathologies [159,160,161]. The microbiome is vital in human homeostasis, especially in relation to metabolism [162]. Several researchers have reported a relation between metabolite profile alteration with anomalous-gut microbiota of patients with various diseases such as metabolic liver disease, inflammatory bowel disease, and metabolic disorders such as malnutrition and obesity [163,164]. Scientific and medical communities have accumulated evidence on the strong bidirectional connection between intestinal microbiota’s metabolites with various diseases as shown in Figure 7. Table 2 summarized the nanomaterial-based platforms for gut metabolite detection.

#### 2.2.1. Trimethylamine-N-oxide (TMAO)

There is substantial evidence that Trimethylamine-N-oxide (TMAO) is a crucial metabolite of gut bacteria. Gut microbiota degrades nutrients rich in trimethylamine (TMA)-containing substances such as choline, carnitine, and lecithin to produce TMAO [166]. Recently, researchers have linked TMAO to an elevated risk of developing complex illnesses such as CVDs, colorectal cancer, chronic kidney diseases, diabetes, obesity [167,168], acute coronary syndrome [169], atherosclerosis, pneumonia, etc., [170]. As a result, TMAO is developing as a critical prognostic and diagnostic biomarker that establishes its biological activities in human health and disease, and its analytical monitoring is critical in health management.

Lakshmi et al. [15] fabricated an electrochemical sensor based on molecularly imprinted polymer (MIP) for selective and sensitive TMAO detection in bodily fluids such as urine [Figure 8i]. The MIP design was centered on polypyrrole (PPy) that was made using a chemical oxidation polymerization technique in the presence and absence of TMAO. The detection response was measured using differential pulse voltammetry (DPV), which indicated a peak current reduction as TMAO concentrations were increased having sensitivity of 2.47 μA mL ppm^−1^ cm^−2^ and a detection range of 1–15 ppm. In another article, using a polyallylamine hydrochloride-capped manganese dioxide (PAH@MnO_2_) nanozyme and a plasma absorption pad/separation pad, a new “color-switch” approach was designed by Chang et al. for the quantitative determination of TMAO directly in whole blood without any pre-treatment step [Figure 8ii]. Through quantitative investigation, the limit of quantitation (LOQ) for TMAO of <6.7 μM was achieved with wide linearity ranging from 15.6–500 μM as proof of concept [171]. Waffo et al. [172] also fabricated an amperometric TMAO biosensor in which glucose oxidase (GOD), TMAO reductase (TorA), and catalase (Cat) were attached to the surface of the electrode, allowing enzymatic TMAO reduction measurements, as shown in Figure 8iii. The sensor depicted linearity for concentrations of TMAO that range between 2–15 mM, having sensitivity 2.75 1.7 μA/mM, 33 s of response time, and 3 weeks stability.

Furthermore, Yi et al. [173] developed a unique TMAO detection technique based on microbial electrochemical technology that directly converts TMAO concentration into electrical signals [Figure 8iv]. This approach demonstrated broad linearity from 0–250 μM, low LOD (5.96 M), and high sensitivity (23.92 μA/mM). Furthermore, the determination procedure was completed in 600 s, having a 90% accuracy rate in actual serum.

#### 2.2.2. Other Gut Metabolites

Among metabolites made by the microbiome, Indole and its derivatives, such as indole-3-propionic acid, indole-3-acetic, indoxyl-3-sulphate, and indole-3-aldehyde, are present in the gut [174]. Indole derivatives go to the blood through intestinal absorption, and then they go to the liver through the blood. In the liver, indole derivatives are transformed into indoxyl sulphate by sequential hydroxylation and sulfation by hepatic cytochrome oxidases CYP 2E1 and SULT1A1, respectively [175]. Indole-3-aldehyde (I3A) may show vast health consequences [176], including gut barrier function regulation, T cells [177], inflammation of the skin [178], inflammation of the central nervous system [179], and inflammation related to aging. Wang et al. [180] fabricated an *E. coli* strain having a single plasmid harboring a chimeric two-component device that determines I3A at 0.1 to 10 μM [Figure 9i]. This novel I3A biosensor enables the indole metabolites detection formed at distinct host–microbe connections and adds innovative components to synthetic biology uses.

Filik et al. [181] developed a graphene-screen-printed electrode-based (GR-SPE) sensor and its electro-activity in the presence of uramic toxin indoxyl sulphate (Inds S) was explored by cyclic and square-wave voltammetry. The output shows that the electrode exposed excellent electro-activity to uramic toxin Inds S, and the oxidation process of Inds S is irreversible and pH dependent. The calibration curve shows the linearity with Inds s concentration varies from 0.5–80 μM with a lower limit of detection 0.064 μM. This GR-SPE shows satisfactory results in human serum and urine samples and can be a promising tool for the routine sensing applications. Michaela.et.al. [182] fabricated a carbon composite film electrode and a carbon paste electrode and measured their electrochemical behavior in the presence of Inds S. The developed sensors were capable of quantifying the limit of Inds S as 0.72 μmol L^−1^ for the carbon composite film electrode and 1.7 μmol L^−1^ for the carbon paste electrode.

TMA is a distinctive odor produced by the fish degradation after its death that results in degradation (TMAO) after enzymatic actions. TMA is practically non-existent in live marine fish, although TMAO is abundant. TMA concentration rises after death, and its detection has been described as an effective indicator of the freshness of fish [183,184]. Diallo et al. [184] developed a field effect transistor-based enzymatic biosensor focused on the TMA determination in the context of seafood freshness evaluation. The device was constructed using a membrane of an enzyme comprising flavin-containing monooxygenase three (FMO_3_) cross-linked with bovine serum albumin (BSA) which was immobilized on silicon nitride. TMA biosensing sensitivity was established in the 0–8 ppm range, as shown in Figure 9ii.

Further, an amperometric and impedimetric biosensor based on the substitution of conducting polypyrrole with ferrocenyl has been fabricated for detecting TMA by Bourigua et al. [184] as shown in Figure 9iii. The biosensor fabrication was characterized by impedance, CV, and FT-IR measurements. Impedimetric and amperometric responses were examined as a function of the concentration of TMA ranging from 0.4 μg mL^−1^ to 80 μg mL^−1^. The impedance spectroscopy calibration curve demonstrates excellent sensitivity with a dynamic detection range that is stable for 16 days. Moreover, Mitsubayashi et al. constructed a biosensor by immobilizing FMO_3_ onto a dissolved oxygen electrode area for TMA detection [Figure 9iv]. The FMO_3_ attached biosensor with flow injection analysis (FIA) shows good linearity with high repeatability for TMA samples that range from 1.0 to 50.0 mmol L^−1^ [185].
Figure 9(**i**) Schematic illustration of the design of an I3A biosensor and detection mechanism of I3A [Adapted with permission from ref. [180], copyright 2021 ACS]; (**ii**) schematic structure of En-OFETs device [Adapted with permission from ref. [183], copyright 2009 Elsevier]; (**iii**) illustration of the chemical process in the biosensor production based on functionalized polypyrrole [Adapted with permission from ref. [184], copyright 2011 Elsevier]; and (**iv**) the experimental setup for the flow injection analysis with the FMO attached biosensor is depicted schematically [Adapted with permission from ref. [185], copyright 2004 Elsevier].
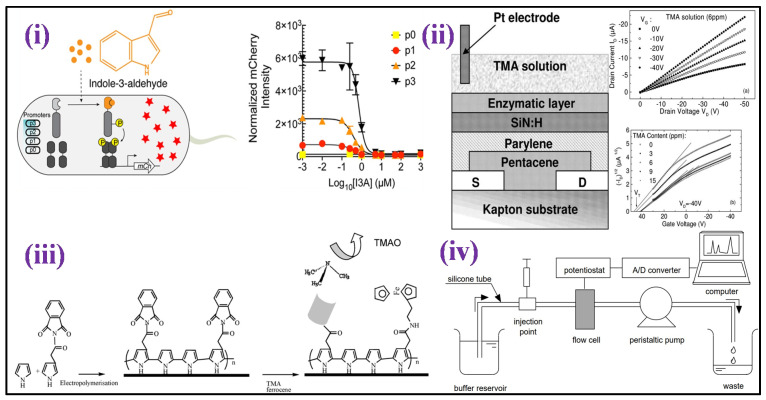

biosensors-12-00733-t002_Table 2Table 2Nanomaterial-based biosensing platform towards clinically relevant gut metabolites detection.MetaboliteNanomaterial Based Biosensing PlatformElectrochemical TechniqueLimit of Detection (LOD)Linear Detection Range (LDR)Real SampleRef.TMAOMIP/ITODPV1 ppm1–15 ppmUrine[15]PAH@MnO_2_-<6.7 μM15.6 to 500 μMUrine[171]TorA/GOD/CatAmperometry10 µM2 µM–15 mMHuman serum[172]S.loihica PV-4Chronoamperometry5.96 lM0 to 250 µMReal serum[173]Indoxyl sulfate (IS)GR-SPESWV0.064 μM0.5–80 μMHuman serum and urine[181]Carbon composite film electrodeVoltammetric0.72 μmol L^−1^-Urine[182]Trimethylamine (TMA)Organic field effect transistors (OFETs)--0–8 ppmMarinefishes and seafood[183]Poly(Py-FMO_3_-ferrocene-co-pyEIS0.4 g mL^−1^0.4–80 gmL^−1^Fish extract[184]FMO_3_ immobilized biosensor--1.0–50.0 mmol L^−1^Fish-extract[185]


## 3. Summary and Future Directions

In the past few decades, electrochemical sensors have gained many researchers’ attention due to their excellent analytical properties and as promising tools for clinical diagnostics. All the electrochemical analyses described in this review carry out quick, sensitive, specific, and affordable investigations of clinically significant human and gut molecules without including pre-concentration steps [Figure 10]. Yet, these methods go through several issues that must be discussed before implementing them for diagnostic purposes. Moreover, the developed biosensors can continuously monitor the target molecules in real time, so there are no requirements for regular sample collection, showing innovative paths for fast and simple testing. Moreover, the construction of biosensing platforms using nanoscale transducers requires new techniques which surpass mass-transfer limits by controlling the motion of the nanoscale fluid. The utilization of nanomaterials for the development of biosensors deal with the biocompatibility and toxicity issues that occur due to their very small size. The stability of the nanomaterials is also a point of consideration because with time the nanomaterials agglomerate, and increase in size, and may change their electro-activity which further changes the performance of the sensors when developed. Apart from the nanomaterial limitations, electrochemical biosensors deal with a stability issue due to the use of biomolecules as a reorganization element. Biomolecules have less shelf-life, thermal stability, and pH stability. Biomolecules such as antibody denature at room temperature which may influence the performance of the biosensor and the diagnosis as well.

Although some improvement has been made in nanomaterial-based electrochemical sensing, more study/analysis is needed. In the future, genetic-profiling-based research would likely increase sensor demand, having enhanced multiplexing capabilities and dealing with multiple biomarkers in a minimum period of time. Moreover, synchronized determination of several analytes needs to have no or little cross-reaction among the target and receptor biomolecules. Therefore, the performance of electrochemical biosensors is based on the selection of receptors and the rational design of the nucleic acid probe. The modification of electrochemical sensors with advanced automated devices can lead to no human interference, i.e., widely available for point-of-care (POC) devices. The conventional techniques described here are primarily involved in the assay automation of crucial elements, such as sample preparation, yet still contain manual phases. On the other hand, the complete automation in electrochemical-based assays generates truthful results without the involvement of skilled persons.

These techniques are good enough in the lab, but to integrate them into clinical governance, artificial intelligence (AI) and Internet of medical things (IoMT) must be integrated with the biosensing concept [186]. Digital healthcare allows automated detection tools and integrated cloud products for traditional healthcare monitoring. This digital transformation improves access to quality healthcare for patients, clinicians, and remote communities. Ultrasound, thermometers, glucometers, and ECG readers are point-of-care (POC) devices with Internet access and cloud storage. These advances are vital for improving healthcare by modifying insulin levels and linking patients with clinicians. IoMT-enabled gadgets transform home healthcare. An intelligent home medicine dispensing system uploads the patient’s medical history to the cloud. It reminds doctors and patients to take medication and warns clinicians when people do not. Technological innovation, industrial adaptability, and urbanisation increase healthcare system demands [187,188].

In summary, the most important challenges should be discussed before clinical use approval of electrochemical sensors. Nevertheless, the extraordinary hard work performed by the researchers’ society allows for tremendous optimism that this approach will be proficient in the future. In addition, the present review is focused on the electrochemical-based biosensors for the detection of various metabolites, but several other sensors/biosensors have been developed based on optical, fluorescence, and colorimetric approaches for metabolite detection. So, there is an opportunity for researchers to summarize the optical, fluorescence, and colorimetric-based work in future.

## 4. Conclusions

Processed foods, which are prevalent in Western diets, of - may contribute significantly to the increase of acquired metabolic disorders in developed nations. The requirement for high-performance detection equipment has risen in response to the sudden progress in detecting biomarkers that are indicators of disease. In conclusion, clinically relevant human and gut metabolites are extremely important in assessing biofluids. Electrochemical techniques in clinics could provide a viable solution to the rising demand for quick, selective, low-cost, and completely automated biomarker analysis. The advancement and progress of electrochemical sensors for clinically relevant gastrointestinal metabolites and biomolecules are discussed in this review article. Yet, there are still issues with analyte selectivity and determining the accurate concentration of unbound vs. bounded analytes. Evaluating the response of biosensors for bounded and unbounded analytes in buffer and the influence of pH on the biosensor response could help in the systematic discovery of optimal sensing parameters and alleviate the challenges of monitoring very low concentrations in aqueous fluids. Furthermore, we recommend that complex fluid-sensing abilities be rigorously tested and reported in accordance with the concentrations encountered in samples such as diluted or undiluted biofluids.

## Figures and Tables

**Figure 6 biosensors-12-00733-f006:**
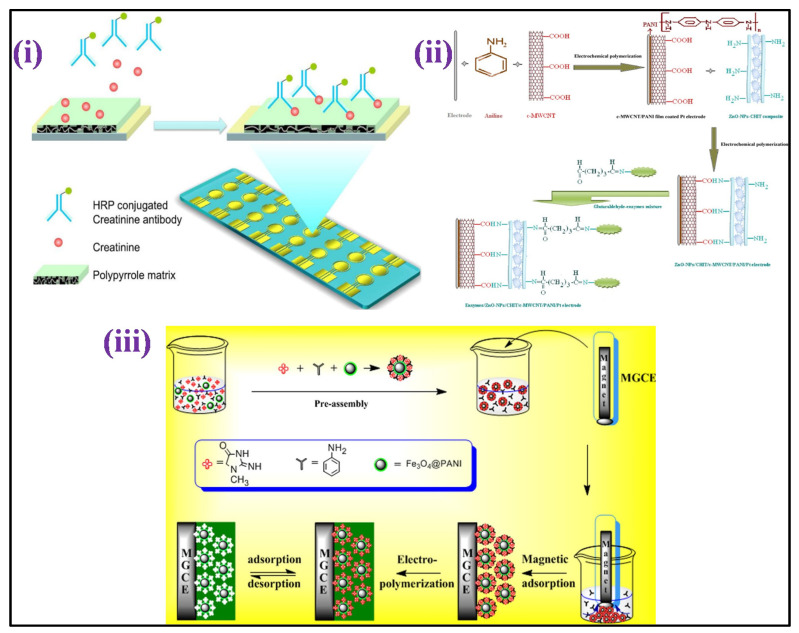
(**i**) Scheme depicting the conducting polymer-electrochemical sensor for creatinine determination in serum [Adapted with permission from ref. [154], Copyright 2012 ACS]; (**ii**) schematic sketch showing chemical reaction involving the development of enzymes/ZnO-NPs/CHIT/c-MWCNT/PANI/Pt hybrid electrode [Adapted with permission from ref. [152], Copyright 2011 Elsevier]; and (**iii**) schematic illustration describing the MIES synthesis for creatinine detection [Adapted with permission from ref. [153], Copyright 2014 Elsevier].

**Figure 7 biosensors-12-00733-f007:**
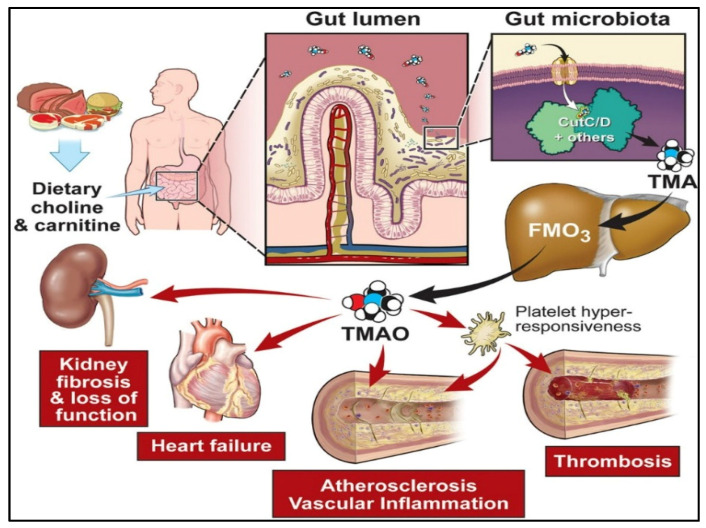
Precursors of diet such as carnitine and choline, metabolized by the gut microbiota into trimethylamine (TMA) through specific genes, including the gut members (choline utilization) gene cluster C/D. [Adapted with permission from ref. [165], copyright 2020 Am Heart Assoc].

**Figure 8 biosensors-12-00733-f008:**
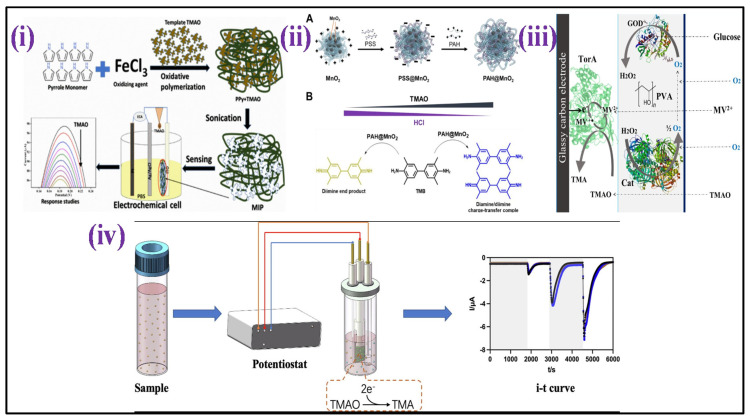
Development process of nanomaterial−based biosensors for TMAO detection: (**i**) scheme depicts the MIP synthesis and the development of an MIP/ITO electrode for detection of TMAO [Adapted with permission from ref. [15], copyright 2021 Springer Nature]; (**ii**) illustration of PAH@MnO_2_ synthesis and colorimetric TMAO determination principle based on proton deposition of HCl by TMAO [Adapted with permission from ref. [171], copyright 2021 MDPI]; (**iii**) TMAO biosensor with various components [Adapted with permission from ref. [172], copyright 2021 MDPI]; and (**iv**) schematic sketch of microbial electrochemical technology established for the TMAO detection [Adapted with permission from ref. [173], copyright 2022 Elsevier].

**Figure 10 biosensors-12-00733-f010:**
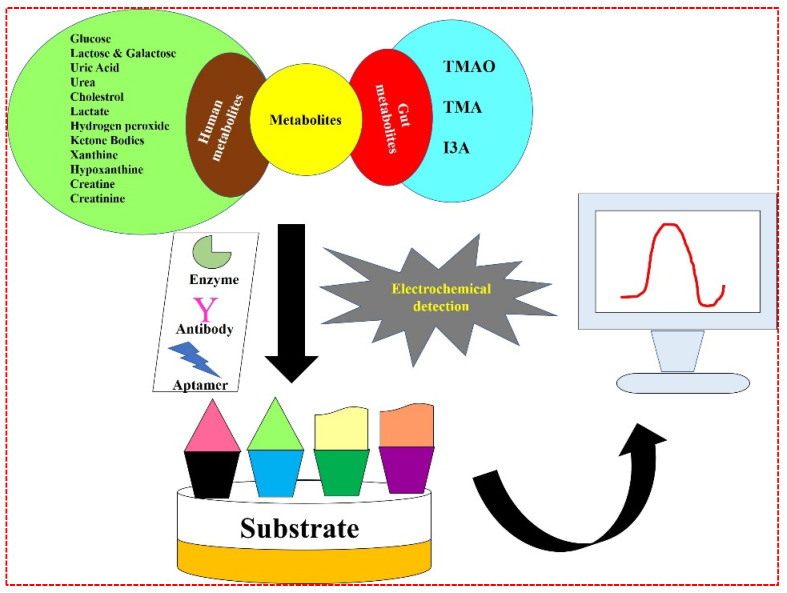
Schematic illustration of all the human and gut metabolites showing electrochemical detection approach.

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
