# Peer review of "Nanomaterial-Based Electrochemical Nanodiagnostics for Human and Gut Metabolites Diagnostics: Recent Advances and Challenges"

_biosensors, 2022, doi:10.3390/bios12090733_

Round 1

Reviewer 1 Report

 Yadav et al., have done a literature survey on the recent advances and challenges in use of nanomaterial in gut metabolites diagnostics. This review is focused on electrochemical biosensing. This review paper has been organised very well and could be of an interest to a broad range of readers. However, there is just a few recommendations be considered before suggesting for publication.

-       It is recommended that the authors expand on the section explaining non-electrochemical techniques or providing more references for readers. “Various conventional and non-electrochemical techniques such as chromatography, raman spectroscopy, surface-enhanced raman scattering (SERS) spectroscopy, and electrochemistry have been developed to analyze metabolites in biological samples [16].” doi.org/10.1021/acsami.8b10590 or doi.org/10.1002/anie.201611243

-       Could the authors summarise different types of the NPs that have been used to detection metabolic analytes mentioned in this review paper?

Author Response

“ANSWER TO REVIEWER’S COMMENTS”

Ms. Ref. No.: biosensors-1873513

Title: "Nanomaterial-based Electrochemical Nanodiagnostics for Human and Gut
Metabolites Diagnostics: Recent Advances and Challenges"

Journal: Biosensors

Dear Editor

We are pleased to submit the revised version of " biosensors-1873513" entitled “Nanomaterial-based Electrochemical Nanodiagnostics for Human and Gut Metabolites Diagnostics: Recent Advances and Challenges”. We appreciate the constructive criticisms of the Editor and the reviewers. We hope that “biosensors” will consider a revised version of the manuscript. We have improved the paper in response to the extensive and insightful 'reviewers' comments. Furthermore, we have rewritten sections of the manuscript, and we sincerely hope that this complies with the 'referee's remarks. We will respond to the comments point-counterpoint, and we have addressed each of their concerns as outlined below. The changes have been highlighted in red in the manuscript.

Here is a point-by-point response to the reviewers’ comments:

REVEIWER 1

Comments and Suggestions for Authors

 Yadav et al., have done a literature survey on the recent advances and challenges in use of nanomaterial in gut metabolites diagnostics. This review is focused on electrochemical biosensing. This review paper has been organised very well and could be of an interest to a broad range of readers. However, there is just a few recommendations be considered before suggesting for publication.

-       It is recommended that the authors expand on the section explaining non-electrochemical techniques or providing more references for readers. “Various conventional and non-electrochemical techniques such as chromatography, raman spectroscopy, surface-enhanced raman scattering (SERS) spectroscopy, and electrochemistry have been developed to analyze metabolites in biological samples [16].” doi.org/10.1021/acsami.8b10590 or doi.org/10.1002/anie.201611243

Response: Thanks for the query. We have rewritten and expanded the section explaining non-electrochemical techniques and have provided two more references for readers as given below-

“SERS is a viable technology to recognize biological and chemical components through their distinct vibrational fingerprints. It opens a new way for biomedical analysis, such as Point-of-care analysis, precision component-based imaging, biomolecular detection, and cancer diagnostics. SERS biosensing at the molecular level has been limited by noble metals and fixed probes [1-2].”

The same has been incorporated in the revised manuscript.

-       Could the authors summarise different types of the NPs that have been used to detection metabolic analytes mentioned in this review paper

Response: Thanks for the comments. We have summarized all the different nanomaterials based biosensing platform mentioned in the manuscript in a tabular form as given below-

Table 1. Nanomaterial based biosensing platform towards clinically relevant human metabolites detection.

Metabolite

Nanomaterial based biosensing platform

Electrochemical technique

Limit of detection (LOD)

Linear detection range (LDR)

Real sample

Ref.

Glucose

Teflon-coated Pt/Ir wire

Amperometry

1.5 μM

5–800 μM

Tear

[46]

µPEDs

Chronoamperometric

1.0 ppb

0.2-22.6 mM

Urine

[47]

PVA-CuO/ITO

Capacitance

-

0.5 - 20 mM

-

[49]

uMED

Chronoamperometry

-

50–500 mgdL-1

Blood

[51]

Con A/Au

Capacitance

1.0 × 10−6 M

1.0 × 10−6 to 1.0 × 10−2 M

-

[54]

GOx/CeO2–TiO2/ITO

DPV

10 mgdL-1

0.56–22.2 mM

-

[56]

GOx/NS-PANI/ITO

DPV

2.1 mM

up to 400 mgdL− 1

-

[57]

GOx/CeO2/Pt bio-

DPV

1.01 mM

25–300 mgdL-1

-

[58]

ITO/PB/(PEI/PVS)1(PEI/β-Gal)30

Amperometry

1.13 mmol L–1

1.13 mmol L–1

Milk

[63]

Au-MPA-[MWCNT-P(AMB-A)-PDA]/CDH

Amperometry

-

0-30 mM

-

[64]

P3HT/SA/b-gal/GaO

Amperometry

-

1–6 gdL-1

[65]

Galactose

P3HT/SA/ITO

Amperometry

-

1–4 gdL-1

Milk

[66]

ITOP3HT/SA/GaO

Amperometry

-

0.05-0.5 g galactoseL-1

Blood

[67]

Uric acid

Naf/UOx/Fc/GCE

CV and DPV

230 nM

500 nM to 600 μM

Blood

[72]

BDD

CV

1 mM

250−1250 μM

Urine

[73]

S-Au electrode

Amperometry

0.4 μM

2.5 μM to 5 mM

urine

[74]

GCE/AuNp@cysteamine/PAMAM

CV

34.5 nM

-

Blood

[75]

Uricase/MWCNT /PANI/ITO

CV

5 μM

0.005–0.6 mM

Serum

[76]

Urea

CS-rGO/Con A

EIS

-           

1-7 mM

Serum

[82]

Urs-GLDH/MLG/ITO

CV

0.6 mM

1.7-16.7 mM

-

[83]

GA-NC/ITO

CV

-

2-20 and 0.1-2 mM

-

[84]

CdS QDs–MIPs/Au

DPV

1.0 × 10−12 M

5.0 × 10−12- 7.0 × 10−8 M

Serum

[86]

Urs-GLDH/TiO2-ZrO2/ITO

CV

0.44 mM

5–100 mgdL-1

-

[87]

Urs-GLDH/GOS/ITO

CV

2.1 mM

3.3-19.9 mM

-

[88]

Urs-GLDH/CDT/Au

CV

9 mgdL-1

10-00 mgdL-1

-

[89]

Urs-GLDH/ ZrO2/Au

CV

5 mgdL−1

5-100 mgdL−1

-

[90]

Cholesterol

ChEt–ChOx/PPY–MWCNT/PTS/ITO

DPV

0.04 mML-1

4 × 10−4-6.5 × 10−3 ML-1

Serum

[93]

ChEt-ChOx/4-ATP/Au

CV

1.34 mM and 1.06 mM

25-400 mgdL−1

-

[94]

ChOx/nan-NiO-CHIT/ITO

CV

43.4 mgdL−1

10–400 mgdL−1

-

[95]

ChOx/Glu/PANI-NT/ITO

LSV

1.18 mM

25-500 mgdL−1

-

[96]

ChOx/PANI-CMC/ITO

1.31 mM

0.5-22 mM

-

[97]

Lactate

Pt electrode

Amperometry

0.8 μM

2 to ∼1000 μM

Whole blood

[110]

Fe3O4/MWCNT/LDH/NAD+/GC

DPV

5 μM

50–500 μM

Serum

[111]

ElecFET

Amperometry

-

1-6 mM

-

[114]

sol-gel/PANI/LDH

Amperometry

-

1 mM-4 mM

-

[115]

PPY–PVS–LDH

Amperometry

1×10−4 M

0.5-6 mM

-

[117]

Hydrogen peroxide

HRP-PANI-ClO4-/ITO

CV

1.984 mM

3-136 mM

-

[120]

HRP/PANI-CeO2/ITO

CV

50 mM

50–500 mM

-

[121]

HRP/NanoCeO2/ITO

CV

0.5 μM

1.0–170 μM

-

[122]

MP-11/MWCNTs–BC

CV

0.1 µM

0.1–257.6 µM

-

[127]

Au−TiO2/Cys

CV

2 nM

10−9 M-10−2 M

-

[129]

Mb/CeO2/ITO

CV

0.6 μM

0.2-5 mM

-

[132]

Ketone bodies

SWCNT-modified SPCE

CV

80 μM

0.1-2 mM

Serum

[140]

ZnO NPs

I–V

68 μM

130 μM-1330 mM

-

[141]

Xanthine

XO-CD/pAuNP/SWNT/GCE

CV

40 nM

50 nM-9.5 μM

-

[144]

PAP/RGO/GC

LSV

0.5 μM

1.0-120μM

-

[145]

EPG/XDH

CV

2.5×10−10 M

1.0 ×10−5 −1.8 × 10−3 M

-

[149]

Creatinine

ZnO-NPs/CHIT/c-MWCNT/PANI/Pt

EIS

0.5 μM

10-650 μM

Blood

[152]

Fe3O4@PANI NPs

DPV

0.35 nmolL-1

2.0×108 – 1.0× 106 molL-1

Urine and plasma

[153]

Conducting polymer

Amperometric

0.46 mgdL-1

0 mgdL-1-11.33 mgdL-1

Blood

[154]

EPPG

CV

0.27 mM

7.5–11.5 mM

Urine

[155]

Creatine

Fe3O4-CPEE

EIS

2.0×10-7 molL-1

2.0×10-7 molL-1 to 3.8×10-6 molL-1

Commercial creatine powder

[151]

Table 2: Nanomaterial based biosensing platform towards clinically relevant gut metabolites detection.

Metabolite

Nanomaterial based biosensing platform

Electrochemical technique

Limit of detection (LOD)

Linear detection range (LDR)

Real sample

Ref.

TMAO

MIP/ITO

DPV

1 ppm

1–15 ppm

Urine

[15]

PAH@MnO2

-

< 6.7 μM

15.6 to 500 μM

Urine

171

TorA/GOD/Cat

Amperometry

10 µM

2 µM -15 mM

Human serum

172

S.loihica PV-4

Chronoamperometry

5.96 lM

0 to 250 µM

Real serum

173

Indoxyl sulfate (IS)

GR-SPE

SWV

0.064 μM

0.5–80 μM

Human serum and urine

181

Carbon composite film electrode

Voltammetric

0.72 μmol L-1

-

Urine

182

Trimethylamine (TMA)

Organic field effect transistors (OFETs)

-

-

0–8 ppm

Marine

fishes and seafood

183

Poly(Py-FMO3- ferrocene-co-py

EIS

0.4 g mL−1

0.4 gmL−1–80 gmL−1

Fish extract

184

FMO3 immobilized biosensor

-

-

1.0- 50.0 mmolL-1

Fish-extract

185

The same has been incorporated in the revised manuscript.

REFERENCES

  1. Keshavarz M, Tan B, Venkatakrishnan K. Label-free SERS quantum semiconductor probe for molecular-level and in vitro cellular detection: a noble-metal-free methodology. ACS applied materials & interfaces. 2018 Sep 21;10(41):34886-904.
  2. Morla-Folch J, Gisbert‐Quilis P, Masetti M, Garcia‐Rico E, Alvarez‐Puebla RA, Guerrini L. Conformational SERS Classification of K‐Ras Point Mutations for Cancer Diagnostics. Angewandte Chemie. 2017 Feb 20;129(9):2421-5.

Reviewer 2 Report

Yadav et al. proposed the review article demonstrating the nanomaterial-based diagnosis of human and gut metabolites.

·       Review article lacks crucial information like development of several nanobiosensor like optical, electrochemical, fluorescence, colorimetric etc.

·       All the figures are adapted from other articles, author should provide some original figure as well.

·       Please insert a section describing the limitations of nano-based diagnosis also.

·       Authors should provide table comparing the several nanomaterial-based methods for detection

·       Most of the reference used are more than 5-year-old, please update the list of references.

·       In Biosensors for gut metabolites detection section, please add other metabolites also

Author Response

“ANSWER TO REVIEWER’S COMMENTS”

Ms. Ref. No.: biosensors-1873513

Title: "Nanomaterial-based Electrochemical Nanodiagnostics for Human Gut and other
Metabolites Diagnostics: Recent Advances and Challenges"

Journal: Biosensors

Dear Editor

We are pleased to submit the revised version of " biosensors-1873513" entitled “Nanomaterial-based Electrochemical Nanodiagnostics for Human and Gut Metabolites Diagnostics: Recent Advances and Challenges”. We appreciate the constructive criticisms of the Editor and the reviewers. We hope that “biosensors” will consider a revised version of the manuscript. We have improved the paper in response to the extensive and insightful 'reviewers' comments. Furthermore, we have rewritten sections of the manuscript, and we sincerely hope that this complies with the 'referee's remarks. We will respond to the comments point-counterpoint, and have addressed each of their concerns as outlined below. The changes have been highlighted in red in the manuscript.

Here is a point-by-point response to the reviewers’ comments:

REVEIWER 2

Comments and Suggestions for Authors

Yadav et al. proposed the review article demonstrating the nanomaterial-based diagnosis of human and gut metabolites.

       Review article lacks crucial information like development of several nanobiosensor like optical, electrochemical, fluorescence, colorimetric etc.

Response: Thank you for your valuable suggestions. The present review is focused on the electrochemical based biosensors for the detection of various metabolites. Therefore, in this review, authors have not described the development of optical, fluorescence and colorimetric nanobiosensors. Moreover, several sensors/biosensors have been developed based on optical, fluorescence and colorimetric approach for metabolites detection. So, there is a wide scope for researchers to summarize the optical, fluorescence and colorimetric-based work in future. The same has been incorporated in the revised manuscript. On other hand, the detailed about electrochemical nanobiosensors have been given in the revised manuscript.

    All the figures are adapted from other articles; author should provide some original figure as well.

Response: Thanks for the advice. We have incorporated a figure (Figure 10) illustrating the schematic view of all the metabolites described in the manuscript along with electrochemical detection approach as given below-

Figure 10. Schematic illustration of all the human and gut metabolites showing electrochemical detection approach.

The same has been incorporated in the revised manuscript.

  • Please insert a section describing the limitations of nano-based diagnosis also.

Response: Thank for your advice.  Author has discussed about the limitation of nanomaterials used for the development of electrochemical biosensors in the section 3 under the heading “Summary and future directions” as given below-

“The utilization of nanomaterials for the development of biosensor deal with the biocompatibility and toxicity issues that are occurs due to their very small size. The stability of the nanomaterials is also a point of consideration because with time the nanomaterials agglomerate and increased their size and may change their electro activity which further changes the performance of the sensors when developed. Apart from the nanomaterial’s limitations, electrochemical biosensors deal with the stability issue which come due to the use of biomolecules as a reorganization element. Biomolecules have less shelf-life, thermal stability, and pH stability. The Biomolecules like antibody denature at room temperature which may influenced the performance of the biosensor and diagnosis as well.”

The same has been incorporated in the revised manuscript.

  • Authors should provide table comparing the several nanomaterial-based methods for detection.

Response: Thanks for the suggestion. We have summarized all the different nanomaterials based biosensing platform mentioned in the manuscript in a tabular form as given below-

Table 1. Nanomaterial based biosensing platform towards clinically relevant human metabolites detection.

Metabolite

Nanomaterial based biosensing platform

Electrochemical technique

Limit of detection (LOD)

Linear detection range (LDR)

Real sample

Ref.

Glucose

Teflon-coated Pt/Ir wire

Amperometry

1.5 μM

5–800 μM

Tear

[46]

µPEDs

Chronoamperometric

1.0 ppb

0.2-22.6 mM

Urine

[47]

PVA-CuO/ITO

Capacitance

-

0.5 - 20 mM

-

[49]

uMED

Chronoamperometry

-

50–500 mgdL-1

Blood

[51]

Con A/Au

Capacitance

1.0 × 10−6 M

1.0 × 10−6 to 1.0 × 10−2 M

-

[54]

GOx/CeO2–TiO2/ITO

DPV

10 mgdL-1

0.56–22.2 mM

-

[56]

GOx/NS-PANI/ITO

DPV

2.1 mM

up to 400 mgdL− 1

-

[57]

GOx/CeO2/Pt bio-

DPV

1.01 mM

25–300 mgdL-1

-

[58]

ITO/PB/(PEI/PVS)1(PEI/β-Gal)30

Amperometry

1.13 mmol L–1

1.13 mmol L–1

Milk

[63]

Au-MPA-[MWCNT-P(AMB-A)-PDA]/CDH

Amperometry

-

0-30 mM

-

[64]

P3HT/SA/b-gal/GaO

Amperometry

-

1–6 gdL-1

[65]

Galactose

P3HT/SA/ITO

Amperometry

-

1–4 gdL-1

Milk

[66]

ITOP3HT/SA/GaO

Amperometry

-

0.05-0.5 g galactoseL-1

Blood

[67]

Uric acid

Naf/UOx/Fc/GCE

CV and DPV

230 nM

500 nM to 600 μM

Blood

[72]

BDD

CV

1 mM

250−1250 μM

Urine

[73]

S-Au electrode

Amperometry

0.4 μM

2.5 μM to 5 mM

urine

[74]

GCE/AuNp@cysteamine/PAMAM

CV

34.5 nM

-

Blood

[75]

Uricase/MWCNT /PANI/ITO

CV

5 μM

0.005–0.6 mM

Serum

[76]

Urea

CS-rGO/Con A

EIS

-           

1-7 mM

Serum

[82]

Urs-GLDH/MLG/ITO

CV

0.6 mM

1.7-16.7 mM

-

[83]

GA-NC/ITO

CV

-

2-20 and 0.1-2 mM

-

[84]

CdS QDs–MIPs/Au

DPV

1.0 × 10−12 M

5.0 × 10−12- 7.0 × 10−8 M

Serum

[86]

Urs-GLDH/TiO2-ZrO2/ITO

CV

0.44 mM

5–100 mgdL-1

-

[87]

Urs-GLDH/GOS/ITO

CV

2.1 mM

3.3-19.9 mM

-

[88]

Urs-GLDH/CDT/Au

CV

9 mgdL-1

10-00 mgdL-1

-

[89]

Urs-GLDH/ ZrO2/Au

CV

5 mgdL−1

5-100 mgdL−1

-

[90]

Cholesterol

ChEt–ChOx/PPY–MWCNT/PTS/ITO

DPV

0.04 mML-1

4 × 10−4-6.5 × 10−3 ML-1

Serum

[93]

ChEt-ChOx/4-ATP/Au

CV

1.34 mM and 1.06 mM

25-400 mgdL−1

-

[94]

ChOx/nan-NiO-CHIT/ITO

CV

43.4 mgdL−1

10–400 mgdL−1

-

[95]

ChOx/Glu/PANI-NT/ITO

LSV

1.18 mM

25-500 mgdL−1

-

[96]

ChOx/PANI-CMC/ITO

1.31 mM

0.5-22 mM

-

[97]

Lactate

Pt electrode

Amperometry

0.8 μM

2 to ∼1000 μM

Whole blood

[110]

Fe3O4/MWCNT/LDH/NAD+/GC

DPV

5 μM

50–500 μM

Serum

[111]

ElecFET

Amperometry

-

1-6 mM

-

[114]

sol-gel/PANI/LDH

Amperometry

-

1 mM-4 mM

-

[115]

PPY–PVS–LDH

Amperometry

1×10−4 M

0.5-6 mM

-

[117]

Hydrogen peroxide

HRP-PANI-ClO4-/ITO

CV

1.984 mM

3-136 mM

-

[120]

HRP/PANI-CeO2/ITO

CV

50 mM

50–500 mM

-

[121]

HRP/NanoCeO2/ITO

CV

0.5 μM

1.0–170 μM

-

[122]

MP-11/MWCNTs–BC

CV

0.1 µM

0.1–257.6 µM

-

[127]

Au−TiO2/Cys

CV

2 nM

10−9 M-10−2 M

-

[129]

Mb/CeO2/ITO

CV

0.6 μM

0.2-5 mM

-

[132]

Ketone bodies

SWCNT-modified SPCE

CV

80 μM

0.1-2 mM

Serum

[140]

ZnO NPs

I–V

68 μM

130 μM-1330 mM

-

[141]

Xanthine

XO-CD/pAuNP/SWNT/GCE

CV

40 nM

50 nM-9.5 μM

-

[144]

PAP/RGO/GC

LSV

0.5 μM

1.0-120μM

-

[145]

EPG/XDH

CV

2.5×10−10 M

1.0 ×10−5 −1.8 × 10−3 M

-

[149]

Creatinine

ZnO-NPs/CHIT/c-MWCNT/PANI/Pt

EIS

0.5 μM

10-650 μM

Blood

[152]

Fe3O4@PANI NPs

DPV

0.35 nmolL-1

2.0×108 – 1.0× 106 molL-1

Urine and plasma

[153]

Conducting polymer

Amperometric

0.46 mgdL-1

0 mgdL-1-11.33 mgdL-1

Blood

[154]

EPPG

CV

0.27 mM

7.5–11.5 mM

Urine

[155]

Creatine

Fe3O4-CPEE

EIS

2.0×10-7 molL-1

2.0×10-7 molL-1 to 3.8×10-6 molL-1

Commercial creatine powder

[151]

Table 2: Nanomaterial based biosensing platform towards clinically relevant gut metabolites detection.

Metabolite

Nanomaterial based biosensing platform

Electrochemical technique

Limit of detection (LOD)

Linear detection range (LDR)

Real sample

Ref.

TMAO

MIP/ITO

DPV

1 ppm

1–15 ppm

Urine

[15]

PAH@MnO2

-

< 6.7 μM

15.6 to 500 μM

Urine

171

TorA/GOD/Cat

Amperometry

10 µM

2 µM -15 mM

Human serum

172

S.loihica PV-4

Chronoamperometry

5.96 lM

0 to 250 µM

Real serum

173

Indoxyl sulfate (IS)

GR-SPE

SWV

0.064 μM

0.5–80 μM

Human serum and urine

181

Carbon composite film electrode

Voltammetric

0.72 μmol L-1

-

Urine

182

Trimethylamine (TMA)

Organic field effect transistors (OFETs)

-

-

0–8 ppm

Marine

fishes and seafood

183

Poly(Py-FMO3- ferrocene-co-py

EIS

0.4 g mL−1

0.4 gmL−1–80 gmL−1

Fish extract

184

FMO3 immobilized biosensor

-

-

1.0- 50.0 mmolL-1

Fish-extract

185

The same has been incorporated in the revised manuscript.

     Most of the reference used are more than 5-year-old, please update the list of references.

      Response: Thanks for the advice. We have updated the list of references with current works for metabolites detection in the revised manuscript as “Ref. [38-39],[59-61],[68-69],[77-78],[80-81], [99-100],[108-109],[123-124],[138-139],[146],[156-158]”.

      In Biosensors for gut metabolites detection section, please add other metabolites also

      Response: Thanks for the comment. We have added few more nanomaterial based electrochemical methods for gut metabolites detection as given below-

“Among metabolites made by the microbiome, Indole and its derivatives such as indole-3-propionic acid, indole-3-acetic, indoxyl-3-sulphate, and indole-3-aldehyde are present in the gut [1]. Indole derivatives go to blood through intestinal absorption, and then it goes to the liver through blood. In the liver, indole derivatives are transformed into indoxyl sulphate by sequential hydroxylation and sulfation by hepatic cytochrome oxidases CYP 2E1 and SULT1A1, respectively [2].

Filik et al., 2016 [3] developed a graphene-screen-printed electrode (GR-SPE) based sensor and their electro-activity in the presence of uramic toxin indoxyl sulphate (Inds S) was explored by cyclic and square- wave voltammetry. The output shows that the electrode exposed excellent electro-activity to uramic toxin Inds S and the oxidation process of Inds S is irreversible and pH dependent. The calibration curve shows the linearity with Inds S concentration varies from 0.5-80 μM with lower limit of detection 0.064 μM. This GR-SPE show satisfactory results in human serum and urine samples and can be promising tool for the routine sensing applications. Michaela et.al., 2018 [4] fabricated carbon composite film electrode and carbon paste electrode and measure their electrochemical behaviour in the presence of Inds S. The developed sensors were capable to quantify the limit of Inds S as 0.72 μmol L-1 for the carbon composite film electrode and 1.7 μmol L-1 for the carbon paste electrode.”

The same has been incorporated in the revised manuscript.

Reference

[1] Beaumont, M. et al. The gut microbiota metabolite indole alleviates liver inflammation in mice. FASEB J fj201800544 (2018).

[2] Skye, S. M. & Hazen, S. L. Microbial Modulation of a Uremic Toxin. Cell Host Microbe 20, 691–692 (2016).

[3] Hayati, F., Asiye, A. A. & Sevda, A. Voltammetric Sensing of Uremic Toxin Indoxyl Sulfate Using High Performance Disposable Screen-Printed Graphene Electrode. Current Pharmaceutical Analysis 12, 36–42 (2016).

[4] Bergerova, M., Libansky, M. & Dejmkova, H. Determination of Indoxyl Sulphate on Carbon Film Composite Electrode and Carbon Paste Electrode. Current Analytical Chemistry 14, 530–535 (2018).

Reviewer 3 Report

This article reviews the development of electrochemical sensors for clinically relevant human metabolites such as glucose, lactose, uric acid, urea, cholesterol, etc., as well as gut metabolites such as TMAO, TMA, and indole derivatives. Various sensing techniques are evaluated in terms of their ability to achieve relevant multiplexing, specificity, and sensitivity limits. I believe that modifications is required to maintain the publication's high quality. 

1.     The authors claim to have discussed the most recent literature, but only 4 articles from the year 2022 are considered. These are not adequate, need to include more. 

2.     This amount of review work is insufficient. What are the advantages and limitations of electrochemical sensors over optical sensors. Additionally, authors should discuss more recent and relevant publications. 

3.     Authors should also categorize their work according to the nanomaterials used for sensing in a tabular form. 

4.     What are the reason for discussion of “glucose, lactose, uric acid, urea, cholesterol” only? There are also some more biomolecules such as like, ALT enzyme, ascorbic acid, are responsible for clinically relevant human metabolites. 

5.     Some sensors are not electrochemical. Please verify and add works to the table. 

6.     The proposed review work must be represented by a solid schematic (add as a Figure 1).

7.   Authors should add the full-form of abbreviations. 

Author Response

“ANSWER TO REVIEWER’S COMMENTS”

Ms. Ref. No.: biosensors-1873513

Title: "Nanomaterial-based Electrochemical Nanodiagnostics for Human and Gut
Metabolites Diagnostics: Recent Advances and Challenges"

Journal: Biosensors

Dear Editor

We are pleased to submit the revised version of "biosensors-1873513" entitled “Nanomaterial-based Electrochemical Nanodiagnostics for Human and Gut Metabolites Diagnostics: Recent Advances and Challenges”. We appreciate the constructive criticisms of the Editor and the reviewers. We hope that “biosensors” will consider a revised version of the manuscript. We have improved the paper in response to the extensive and insightful 'reviewers' comments. Furthermore, we have rewritten sections of the manuscript, and we sincerely hope that this complies with the 'referee's remarks. We will respond to the comments point-counterpoint, and we have addressed each of their concerns as outlined below. The changes have been highlighted in red in the manuscript.

Here is a point-by-point response to the reviewers’ comments:

REVIEWER 3

Comments and Suggestions for Authors

This article reviews the development of electrochemical sensors for clinically relevant human metabolites such as glucose, lactose, uric acid, urea, cholesterol, etc., as well as gut metabolites such as TMAO, TMA, and indole derivatives. Various sensing techniques are evaluated in terms of their ability to achieve relevant multiplexing, specificity, and sensitivity limits. I believe that modifications is required to maintain the publication's high quality.

  1. The authors claim to have discussed the most recent literature, but only 4 articles from the year 2022 are considered. These are not adequate, need to include more.

      Response: Thanks for the advice. We have updated the list of references with current works for metabolites detection in the revised manuscript as “Ref. [38-39],[59-61],[68-69],[77-78],[80-81], [99-100],[108-109],[123-124],[138-139],[146],[156-158]”.

  1. This amount of review work is insufficient. What are the advantages and limitations of electrochemical sensors over optical sensors. Additionally, authors should discuss more recent and relevant publications.

Response: Thanks for the query. We have incorporated more recent work related to each metabolite’s detection based on electrochemical approach in the revised manuscript. The changes have been highlighted in red.

Moreover, the present review summarized only the electrochemical based biosensors/sensors for metabolites detection and not the optical sensors/biosensors. We have removed the  optical based biosensors from the revised manuscript. In addition, the advantages and disadvantages of electrochemical biosensors/sensors have been already described in the manuscript under heading of “Summary and future perspective” as given below:-

Summary and Future Directions

In the past few decades, electrochemical sensors have gained many researchers’ attention due to its excellent analytical properties and as promising tools for upcoming clinical diagnostics. All the electrochemical analyses described in this review carry out quick, sensitive, specific, and affordable investigations of clinically significant human and gut molecules without including pre-concentration steps. Yet, these methods go through several issues that must be discussed before implementing these methods for diagnosis purposes. Moreover, the developed biosensors can continuously monitor the target molecules in real time, so there are no requirements for regular sample collection showing innovative paths for fast and simple testing. Moreover, the construction of biosensing platforms using nanoscale transducers required new techniques which overcome the mass-transfer limits by controlling the motion of the nanoscale fluid. The utilization of nanomaterials for the development of biosensor deal with the biocompatibility and toxicity issues that are occurs due to their very small size. The stability of the nanomaterials is also a point of consideration because with time the nanomaterials agglomerate and increased their size and may change their electro activity which further changes the performance of the sensors when developed. Apart from the nanomaterials limitation electrochemical biosensors deal with the stability issue which come due to the use of biomolecules as a reorganization element. Biomolecules have less shelf-life, thermal stability, and pH stability. The biomolecules like antibody denature at room temperature which may influenced the performance of the biosensor and diagnosis as well.

Although some improvement has been made in nanomaterials-based electrochemical sensing, more study/ analysis is still needed. In the future, genetic profiling-based research would likely increase sensor demand, having enhanced multiplexing capabilities and dealing with multiple biomarkers in a minimum period of time. Besides, synchronized determination of several analytes needs no or less cross-reaction among the target and receptors biomolecules. Therefore, the performance of electrochemical biosensors is based on the selection of receptors and the rational design of the nucleic acid probe. The modification of electrochemical sensors with advanced automated devices can lead to no human interference i.e., widely vital for point-of-care (POC) devices. The conventional techniques described here are primarily anxious with the assay automation of crucial elements, like sample preparation, yet still contain manual phases. On the other hand, the complete automation in electrochemical-based assays generates truthful results without the involvement of skilled persons. 

In summary, the most important challenges should be discussed before the clinical use approval of electrochemical sensors. Nevertheless, the extraordinary hard work committed by the researchers’ society allows for tremendous optimism that this purpose would be proficient in the upcoming future ahead.”

Further, we have discussed the more recent and relevant publications in the revised manuscript.

  1. Authors should also categorize their work according to the nanomaterials used for sensing in a tabular form.

Response: Thanks for the comments. We have summarized all the different nanomaterials based biosensing platform mentioned in the manuscript in a tabular form as given below-

Table 1. Nanomaterial based biosensing platform towards clinically relevant human metabolites detection.

Metabolite

Nanomaterial based biosensing platform

Electrochemical technique

Limit of detection (LOD)

Linear detection range (LDR)

Real sample

Ref.

Glucose

Teflon-coated Pt/Ir wire

Amperometry

1.5 μM

5–800 μM

Tear

[46]

µPEDs

Chronoamperometric

1.0 ppb

0.2-22.6 mM

Urine

[47]

PVA-CuO/ITO

Capacitance

-

0.5 - 20 mM

-

[49]

uMED

Chronoamperometry

-

50–500 mgdL-1

Blood

[51]

Con A/Au

Capacitance

1.0 × 10−6 M

1.0 × 10−6 to 1.0 × 10−2 M

-

[54]

GOx/CeO2–TiO2/ITO

DPV

10 mgdL-1

0.56–22.2 mM

-

[56]

GOx/NS-PANI/ITO

DPV

2.1 mM

up to 400 mgdL− 1

-

[57]

GOx/CeO2/Pt bio-

DPV

1.01 mM

25–300 mgdL-1

-

[58]

ITO/PB/(PEI/PVS)1(PEI/β-Gal)30

Amperometry

1.13 mmol L–1

1.13 mmol L–1

Milk

[63]

Au-MPA-[MWCNT-P(AMB-A)-PDA]/CDH

Amperometry

-

0-30 mM

-

[64]

P3HT/SA/b-gal/GaO

Amperometry

-

1–6 gdL-1

[65]

Galactose

P3HT/SA/ITO

Amperometry

-

1–4 gdL-1

Milk

[66]

ITOP3HT/SA/GaO

Amperometry

-

0.05-0.5 g galactoseL-1

Blood

[67]

Uric acid

Naf/UOx/Fc/GCE

CV and DPV

230 nM

500 nM to 600 μM

Blood

[72]

BDD

CV

1 mM

250−1250 μM

Urine

[73]

S-Au electrode

Amperometry

0.4 μM

2.5 μM to 5 mM

urine

[74]

GCE/AuNp@cysteamine/PAMAM

CV

34.5 nM

-

Blood

[75]

Uricase/MWCNT /PANI/ITO

CV

5 μM

0.005–0.6 mM

Serum

[76]

Urea

CS-rGO/Con A

EIS

-           

1-7 mM

Serum

[82]

Urs-GLDH/MLG/ITO

CV

0.6 mM

1.7-16.7 mM

-

[83]

GA-NC/ITO

CV

-

2-20 and 0.1-2 mM

-

[84]

CdS QDs–MIPs/Au

DPV

1.0 × 10−12 M

5.0 × 10−12- 7.0 × 10−8 M

Serum

[86]

Urs-GLDH/TiO2-ZrO2/ITO

CV

0.44 mM

5–100 mgdL-1

-

[87]

Urs-GLDH/GOS/ITO

CV

2.1 mM

3.3-19.9 mM

-

[88]

Urs-GLDH/CDT/Au

CV

9 mgdL-1

10-00 mgdL-1

-

[89]

Urs-GLDH/ ZrO2/Au

CV

5 mgdL−1

5-100 mgdL−1

-

[90]

Cholesterol

ChEt–ChOx/PPY–MWCNT/PTS/ITO

DPV

0.04 mML-1

4 × 10−4-6.5 × 10−3 ML-1

Serum

[93]

ChEt-ChOx/4-ATP/Au

CV

1.34 mM and 1.06 mM

25-400 mgdL−1

-

[94]

ChOx/nan-NiO-CHIT/ITO

CV

43.4 mgdL−1

10–400 mgdL−1

-

[95]

ChOx/Glu/PANI-NT/ITO

LSV

1.18 mM

25-500 mgdL−1

-

[96]

ChOx/PANI-CMC/ITO

1.31 mM

0.5-22 mM

-

[97]

Lactate

Pt electrode

Amperometry

0.8 μM

2 to ∼1000 μM

Whole blood

[110]

Fe3O4/MWCNT/LDH/NAD+/GC

DPV

5 μM

50–500 μM

Serum

[111]

ElecFET

Amperometry

-

1-6 mM

-

[114]

sol-gel/PANI/LDH

Amperometry

-

1 mM-4 mM

-

[115]

PPY–PVS–LDH

Amperometry

1×10−4 M

0.5-6 mM

-

[117]

Hydrogen peroxide

HRP-PANI-ClO4-/ITO

CV

1.984 mM

3-136 mM

-

[120]

HRP/PANI-CeO2/ITO

CV

50 mM

50–500 mM

-

[121]

HRP/NanoCeO2/ITO

CV

0.5 μM

1.0–170 μM

-

[122]

MP-11/MWCNTs–BC

CV

0.1 µM

0.1–257.6 µM

-

[127]

Au−TiO2/Cys

CV

2 nM

10−9 M-10−2 M

-

[129]

Mb/CeO2/ITO

CV

0.6 μM

0.2-5 mM

-

[132]

Ketone bodies

SWCNT-modified SPCE

CV

80 μM

0.1-2 mM

Serum

[140]

ZnO NPs

I–V

68 μM

130 μM-1330 mM

-

[141]

Xanthine

XO-CD/pAuNP/SWNT/GCE

CV

40 nM

50 nM-9.5 μM

-

[144]

PAP/RGO/GC

LSV

0.5 μM

1.0-120μM

-

[145]

EPG/XDH

CV

2.5×10−10 M

1.0 ×10−5 −1.8 × 10−3 M

-

[149]

Creatinine

ZnO-NPs/CHIT/c-MWCNT/PANI/Pt

EIS

0.5 μM

10-650 μM

Blood

[152]

Fe3O4@PANI NPs

DPV

0.35 nmolL-1

2.0×108 – 1.0× 106 molL-1

Urine and plasma

[153]

Conducting polymer

Amperometric

0.46 mgdL-1

0 mgdL-1-11.33 mgdL-1

Blood

[154]

EPPG

CV

0.27 mM

7.5–11.5 mM

Urine

[155]

Creatine

Fe3O4-CPEE

EIS

2.0×10-7 molL-1

2.0×10-7 molL-1 to 3.8×10-6 molL-1

Commercial creatine powder

[151]

Table 2: Nanomaterial based biosensing platform towards clinically relevant gut metabolites detection.

Metabolite

Nanomaterial based biosensing platform

Electrochemical technique

Limit of detection (LOD)

Linear detection range (LDR)

Real sample

Ref.

TMAO

MIP/ITO

DPV

1 ppm

1–15 ppm

Urine

[15]

PAH@MnO2

-

< 6.7 μM

15.6 to 500 μM

Urine

171

TorA/GOD/Cat

Amperometry

10 µM

2 µM -15 mM

Human serum

172

S.loihica PV-4

Chronoamperometry

5.96 lM

0 to 250 µM

Real serum

173

Indoxyl sulfate (IS)

GR-SPE

SWV

0.064 μM

0.5–80 μM

Human serum and urine

181

Carbon composite film electrode

Voltammetric

0.72 μmol L-1

-

Urine

182

Trimethylamine (TMA)

Organic field effect transistors (OFETs)

-

-

0–8 ppm

Marine

fishes and seafood

183

Poly(Py-FMO3- ferrocene-co-py

EIS

0.4 g mL−1

0.4 gmL−1–80 gmL−1

Fish extract

184

FMO3 immobilized biosensor

-

-

1.0- 50.0 mmolL-1

Fish-extract

185

The same has been incorporated in the revised manuscript.

  1. What are the reason for discussion of “glucose, lactose, uric acid, urea, cholesterol” only? There are also some more biomolecules such as like, ALT enzyme, ascorbic acid, are responsible for clinically relevant human metabolites.

Response: Thanks for the query. We agreed with your suggestion. But, the present review belongs to the Special Issue in Honor of Professor Bansi D. Malhotra; From Nanosystems to a Biosensing Prototype for an Efficient Diagnostic. Moreover, we have almost covered all the clinically relevant human and gut metabolites having more clinical significance in present times.

  1. Some sensors are not electrochemical. Please verify and add works to the table.

Response: Thanks for the advice. As the present review focused on the electrochemical sensors/biosensors, therefore all the optical sensors/biosensors have been removed from the revised manuscript.

  1. The proposed review work must be represented by a solid schematic (add as a Figure 1).

Response: Thanks for the suggestion. We have incorporated a figure (Figure 10) illustrating the schematic view of all the metabolites described in the manuscript along with electrochemical detection approach as given below-

Figure 10. Schematic illustration of all the human and gut metabolites showing electrochemical detection approach.

The same has been incorporated in the revised manuscript.

  1. Authors should add the full-form of abbreviations.

Response: Thanks for the advice. We have incorporated the full form of abbreviations in the separate section in the revised manuscript. The details have been given below- 

Abbreviations:

P3HT, poly(3-hexyl thiophene); SA, stearic acid; ITO, indium tin oxide; S-Au electrode, sulfur-adlayer-coated gold; AuNp@cysteamine, cysteamine-capped gold nanoparticles; PAMAM, poly(amidoamine); GCE, Glassy carbon electrodes; BDD, boron doped diamond; Naf, Nafion; UOx, uricase; Fc, ferrocene; MWCNT, multi-walled carbon nanotube; PANI, polyaniline; CS, chitosan; rGO, reduced graphene oxide; Con A, concanavalin A; MLG, Multilayered graphene; GLDH, glutamate dehydrogenase; GA-NC, gelatin organogel-based nanocomposite; Au, gold; MIPs, Molecularly imprinted polymers; TiO2–ZrO2, titania–zirconia; Urs, Urease; ChOx, Cholesterol oxidase; CHIT, chitosan; NiO, nanostructured nickel oxide; PANI-CMC, polyaniline-carboxymethyl cellulose; ElecFET, Electrochemical Field Effect Transistor; LDH, lactate dehydrogenase; PPY-PVS, polypyrrole–polyvinylsulphonate; ClO4-, perchlorate; MP-11, microperoxidase-11; PAP, Poly(o-aminophenol-co-pyrogallol); RGO/GC, reduced graphene oxide/glassy carbon; ZnO-NPs/CHIT/c-MWCNT/PANI, zinc oxide nanoparticles/chitosan/carboxylated multiwall carbonnanotube/polyaniline; EPPG, Edge Plane Pyrolytic Graphite; SERS, Surface enhanced Raman spectroscopy; HPLC, High performance liquid chromatography; SPE, Solid phase extraction; UA, Uric acid; CA, Chronoamperometry; EIS, Electrochemical impedance spectroscopy; DPV, Differential pulse voltammetry; SWV, Square wave voltammetry; LSV, Linear sweep voltammetry; CV, Cyclic voltammetry; GO, Graphene oxide; CNT, Carbon nanotube; SWCNT, Single-walled carbon nanotube; LOD, Limit of detection; PPY, Poly(pyrrole); CD, Cyclodextrin; MCPE, Modified carbon paste electrode; SPCE, Screen printed carbon electrode.
